# Recent Progress on Fullerene-Based Materials: Synthesis, Properties, Modifications, and Photocatalytic Applications

**DOI:** 10.3390/ma13132924

**Published:** 2020-06-30

**Authors:** Sai Yao, Xingzhong Yuan, Longbo Jiang, Ting Xiong, Jin Zhang

**Affiliations:** 1College of Environmental Science and Engineering, Hunan University, Changsha 410082, China; yaosai@hnu.edu.cn (S.Y.); xiongt@hnu.edu.cn (T.X.); zhangjinjodg@gmail.com (J.Z.); 2Key Laboratory of Environmental Biology and Pollution Control (Hunan University), Ministry of Education, Changsha 410082, China

**Keywords:** fullerene, visible-light photocatalysis, synthesis, H_2_ production, wastewater treatment

## Abstract

Solar light is an inexpensive energy source making up for energy shortage and solving serious environmental problems. For efficient utilization of solar energy, photocatalytic materials have attracted extensive attention over the last decades. As zero-dimensional carbon nanomaterials, fullerenes (C_60_, C_70_, etc.) have been extensively investigated for photocatalytic applications. Due to their unique properties, fullerenes can be used with other semiconductors as photocatalyst enhancers, and also as novel photocatalysts after being dispersed on non-semiconductors. This review summarizes fullerene-based materials (including fullerene/semiconductors and fullerene/non-semiconductors) for photocatalytic applications, such as water splitting, Cr (VI) reduction, pollutant degradation and bacterial disinfection. Firstly, the optical and electronic properties of fullerene are presented. Then, recent advances in the synthesis and photocatalytic mechanisms of fullerene-based photocatalysts are summarized. Furthermore, the effective performances of fullerene-based photocatalysts are discussed, mainly concerning photocatalytic H_2_ generation and pollutant removal. Finally, the current challenges and prospects of fullerene-based photocatalysts are proposed. It is expected that this review could bring a better understanding of fullerene-based photocatalysts for water treatment and environmental protection.

## 1. Introduction

There is no denying that both environmental issues and the energy crisis are becoming serious threats to the sustainable development of human society, with the endless consumption of fossil fuels and the irregular discharge of anthropogenic action [1,2]. In order to solve these problems, industrial development must concentrate on clean energy alternatives, which reduce environmental pollution. As a renewable energy source, solar energy has been an intriguing option. Photocatalysts are an effective route to utilize solar energy for various chemical reactions, including photocatalytic pollutant degradation, disinfection, selective organic synthesis, reduction of CO_2_ and H_2_ generation. This is an attractive technology which could effectually utilize solar energy, generate clean production (H_2_) and remediate the environment. Since the photocatalytic performance of TiO_2_ for water splitting was proposed for the first time by Fujishima and Honda in 1970s, much work has been done to study the photocatalytic mechanisms and develop novel photocatalysts [3]. Up to date, numerous appealing photocatalysts have been developed and extensively investigated, such as simple oxides (ZnO) [4], metal chalcogenides (CdS) [5], Ag-based compound (Ag_3_PO_4_) [6], Bi-based compound (BiVO_4_ and Bi_2_MoO_6_) [7,8], MOFs [9] and g-C_3_N_4_ [10]. In addition to novel photocatalysts, cocatalysts such as precious metals (Pt), two-dimensional transition metal sulfides (MoS_2_, WS_2_, etc.) and carbonaceous nanomaterials are also widely developed and applied in the field of photocatalysis [11,12,13].

Since the mid-1990s, carbonaceous nanomaterials have been attracting extensive attention, including fullerene, carbon nanotube (CNT) and graphene [14]. Due to uniquely optical and electrical properties, they have been extensively investigated in photocatalytic applications in the past decades. On one hand, they could enhance the photocatalytic efficiency of other semiconductors after combination. For example, CNTs could induce photocatalytic enhancement via three mechanisms: increasing the surface area, suppressing the recombination of hole(h^+^)-electron(e^−^) pairs and enhancing the adsorption of visible light) [15]. Similar to CNT, graphene covers all three of the mechanisms of photocatalytic enhancement above. On the other hand, carbonaceous nanomaterials display effective photocatalytic performance on their own without combining with other semiconductors and are applied alone as novel photocatalysts in some cases. For example, Luo, et al. [16] proposed a self-photocatalytic activity of multiwalled nanotubes (MWCNTs) in the visible range after highly defective modification. Moreover, modified graphene oxide (GO) with a band gap of 2.4−4.3 eV exhibits effective H_2_ generation ability within light illumination (UV or visible), which alone may be regarded as a next-generation photocatalyst [17,18].

Among carbonaceous nanomaterials, fullerene exhibits appealing performances similar to CNT and graphene in the photocatalytic application. In previous studies, extensive attentions have been devoted to exploring the roles that fullerene plays in the photocatalytic processes. It was proven that fullerene can be used not only as a photocatalytic enhancer for other semiconductors but also as a novel photocatalyst itself, after being dispersed on a non-semiconductor support. This is ascribed to its distinct optical, photophysical and photochemical properties. Fullerene, a carbon allotrope, is a kind of zero-dimensional (0D) nanocarbon material discovered by Kroto et al., and it has a closed-cage spherical structure which consists of five-membered and six-membered rings [19]. It is well established that there are various forms of fullerene, such as C_60_, C_70_, C_76_, C_82_ and C_84_. Among these forms, the C_60_ and C_70_ were more extensively investigated than others. Owing to electron delocalization, fullerenes are used extensively as strong-affinity electron acceptor, and for instance C_60_ is able to reversibly absorb six electrons [20]. The band gap energy (Eg) of solid fullerenes (such as C_60_, C_70_, C_84_ etc.) are from 1.5 to 1.98 eV between the highest occupied molecular orbital (HOMO) and the lowest unoccupied molecular orbital (LUMO) [4]. Ascribed to the narrow Eg, fullerenes have intensive absorption of UV light and moderate but extensive adsorption of visible light, which make them appealing options for photocatalytic application. Additionally, fullerenes have been previously reported as an excellent photosensitizer as well, with a high quantum efficiency around 1.0 [21]. Fullerene solutions can induce photochemical reactive oxygen species (ROS) generation via two pathways. Under light irradiation (UV or visible), single oxygen (^1^O_2_) will be formed in fullerene-toluene solution (pathway II), and superoxide anion radical (O_2_^−•^) and hydroxyl radical (•OH) can be generated in solvent in the presence of electron donors such as ethylenediamine tetraacetic acid (EDTA) and nicotinamide adenine dinucleotide (NADH) [22]. Typically, ROS is a class of active materials that easily induce chemical reaction, which could play an effective role in photocatalytic application. However, easy aggregation is the main obstacle of fullerene in water treatment applications, which suppresses the photoactivity of fullerene. Namely, when dispersed in water, fullerene tends to form nanoscale aggregates (termed nC_60_, nC_70_, etc.) with the quenching of excited states of neighboring fullerene molecules which are brought into close contact via aggregation. For retaining fullerene’s photoactivity in aqueous systems, it is necessary for immobilization of fullerene onto solid supports.

Nowadays, many fullerene-semiconductor materials have been successfully built for photocatalytic applications, such as TiO_2_/C_60_ (C_70_), ZnO/C_60_, CdS/C_60_ and C_3_N_4_/C_60_ (C_70_) [23,24,25]. These photocatalysts have been extensively investigated in photocatalytic pollutant degradation, disinfection and water splitting for H_2_ evolution. Note that fullerene can obviously enhance the photocatalytic efficiency. At the same time, a variety of fullerene-support (non-semiconductor) materials were successfully fabricated and used for photodegradation of organic pollutant, photocatalytic organic synthesis and disinfection, such as silica/C_60_, γ-Al_2_O_3_/C_60_, MCM-41/C_70_ and polysiloxane-supported fullerene derivative [26,27,28]. Apart from high-efficient photocatalytic activity, these photocatalysts not only exhibited more stable than the pristine fullerene in solution but also had superior recyclability.

Previously, Yeh, Cihlář, Chang, Cheng and Teng [13] have reviewed the roles of graphene oxide (GO) in photocatalytic water splitting, which mainly introduces strategies for tuning the electronic structure of GO for photocatalytic water splitting. Gangu, Maddila and Jonnalagadda [15] have reported a review on the MWCNTs mediated semiconducting materials as photocatalysts in water treatment. In another review, Ge, Zhang and Park [14] have discussed recent advances in carbonaceous photocatalysts and the developmental direction for them, such as activated carbon, carbon dots, carbon nanotubes, graphene and fullerene. To our knowledge, no papers have reviewed the fullerene/semiconductor and fullerene/support photocatalysts for wastewater treatment and water splitting. Therefore, the present review provides a comprehensive understanding of fullerene-based photocatalysts, including fullerene/semiconductor photocatalysts and fullerene/support photocatalysts. The optical, photochemical and electronic properties of fullerene are generally presented. Then, recent advances in the synthesis methods and photocatalytic application of fullerene-based photocatalysts are summarized. Meanwhile, the photocatalytic efficiency of these-prepared photocatalysts are discussed in wastewater treatment and water splitting for H_2_ evolution, wherein the mechanisms of the fullerene-based photocatalysts are underlined in detail. In the end, the current challenges and prospects of fullerene-based photocatalysts are proposed.

## 2. Role of Fullerene

### 2.1. Basic Principles of Semiconducting Photocatalysis

In the photocatalytic procedure of semiconductors, there are three main factors, i.e., light resources, photocatalysts and reaction mediums [14]. The photocatalytic process could be initiated only by the light (i.e., UV, infrared and visible light) with energy equal to or over the band gap energy (Eg) of the photocatalyst. Typically, it could be briefly presented as follows. Upon irradiation by light resource, the electrons in the valence band (VB) could be excited to the conduction band (CB) of the photocatalyst and holes leave in VB, resulting in the separation of photogenerated hole-electron pairs. Immediately, most of them recombine with heat generation while a small fraction can transfer to the semiconductor’s surface to induce redox reactions. Generally, for photocatalytic decontamination, the separated holes and electrons of the semiconductor can react with ambient substances (i.e., H_2_O and O_2_) to produce free radicals (i.e., •OH, •O_2_^−^, HO_2_•, H_2_O_2_). Then, the highly oxidative holes and reactive radicals will intensively degrade organic pollutants into small molecules or inorganic materials through addition/substitution reaction and electron transfer between contaminants and free radicals [29]. Through the processes above, the pollution is mitigated or eliminated. Compared with pollutant degradation, the photocatalytic H_2_ generation over semiconductor share some similarities. In detail, the generating processes of the photoinduced charges and formation of partial radicals are identical between pollutant degradation and H_2_ generation. The difference in photocatalytic H_2_ generation is that photoinduced charges react with H^+^ adsorbed on the photocatalyst or in surroundings to produce H_2_ rather than •O_2_^−^ [30]. In this process, additives are usually required to facilitate the efficiency of photocatalytic H_2_ generation, such as hole scavenger and (or) sacrificial donor.

In the past decades, a number of semiconductors were developed for photocatalysis applications, such as TiO_2_, ZnO, CdS, Ag-based semiconductors, Bi-based semiconductors and g-C_3_N_4_. However, many problems limit the photocatalytic efficiency of the current photocatalysts, including insufficient visible light utilization, wide bandgap, rapid recombination of photoinduced holes and electrons and poor stability. Various strategies were proved to be effective in enhancing the photocatalytic activity, such as morphology control, element doping, heterojunction construction and coupling with carbonaceous nanomaterials [10].

### 2.2. The Role of Fullerene in Semiconductor/Fullerene Photocatalysts

For effective semiconductor/fullerene photocatalysts, the introduction of fullerene generally enhances the photocatalytic performance through various aspects as follows. For instance, fullerene could capture electrons from CB of semiconductor due to its high-affinity for electrons, which significantly retard the recombination of photoinduced hole-electron pairs [31,32]. As a result, more separated photogenerated holes and electrons could take part in photocatalytic reaction, increasing photocatalytic efficiency. Then, fullerene could enhance the light absorption (both UV and visible light) because it is an excellent photo-response material (300~700 nm), wherein elevated light energy utilization excites more electrons from VB to CB [33,34]. In respect to photo-response characteristics, it must also be mentioned that fullerene could not shift the adsorption edge of pristine photocatalyst unless the introduction of fullerene changed the structure of semiconductor [35,36,37]. In other words, if the fullerene does not change the crystalline structure of semiconductor in the synthesis procedure, which was typically affected by a stronger bonding force other than simply physically blending, it could not change the conduction band (Eg) at all. Undoubtedly, the introduction of fullerene could change the specific surface area BET (Brunauer–Emmett–Teller), while it does not always present the same results of change trend. In previous studies, it increased or decreased the BET of the photocatalyst depending on the specific situation [38,39,40]. So, it was controversial that the fullerene enhances the BET of photocatalyst to contribute to high adsorption ability for reactant.

Fullerene is virtually insoluble in water, but soluble in nonpolar organic solvents, such as toluene and 1,2-dichlorobenzen [41]. Hence, it performs a good stability in water solution. After dissolved in nonpolar solvent, fullerene could be coupled with semiconductor to form stable semiconductor/fullerene composite by various methods, wherein no fullerene is leached out even after the as-prepared composite is repeatedly rinsed with the aforementioned solvent. Additionally, fullerene exhibits superior potentials for stabilizing semiconductors which deeply suffers photocorrosion, such as ZnO, CdS and Ag_3_PO_4_. For example, a significant increase of stabilization was observed for ZnO/C_60_ nanocomposite in previous studies [42]. The photogenerated holes of ZnO could easily react with surface oxygen atoms during the photocatalysis process, leading to fast decline of photocatalytic activity. When C_60_ was covered on ZnO, the activity of surface oxygen atoms was effectively reduced so that photocorrosion effect was effectively inhibited. Similarly, high stabilization of AgPO_4_/C_60_ was also observed, because C_60_ could obviously suppress the transform of Ag^+^ into Ag of bare Ag_3_PO_4_ composite [6]. Moreover, Cai, et al. [43] fabricated CdS/C_60_ nanocomposite in which C_60_ has shown effectively inhibition of photocorrosion of CdS. In order to estimate the stability of the as-prepared sample, the released Cd^2+^ concentration was determined in remaining solution after three cycles for rhodamine (RhB) degradation. The Cd^2+^ concentration was 381.3 µg/L in solution with naked CdS while it was 51.9 µg/L in solution with 0.4C_60_/CdS nanocomposite, and the former was 7.3-times of the later. This means that C_60_ could effectively inhibit the photocorrosion to enhance the stability of CdS.

## 3. Synthesis of Semiconductor/Fullerene Photocatalysts

A number of fullerene/semiconductors (TiO_2_, ZnO, CdS, C_3_N_4_, etc.) have been fabricated for photocatalytic wastewater treatment and water splitting. It is unquestionable that synthetic process plays an important role in determining the size, morphology and physicochemical characteristic of a photocatalyst. Fullerene/semiconductor photocatalysts can be constructed via a series of synthetic methods, including simple adsorption, hydrothermal/solvothermal synthesis, sol-gel procedure and mechanical ball-milling. These synthetic methods are summarized and illustrated briefly as follows.

### 3.1. Simple Adsorption Method

Simple adsorption method has been extensively used to fabricate semiconductor/fullerene nanocomposites. This procedure is cost-effective without complicated external condition. It is well established that pure fullerene (C_60_ or C_70_) is extremely hydrophobic but dissolves in some organic solvents (benzene, toluene, 1,2-dichlorobenzen etc.), which are mainly used to dissolve fullerene for preparing fullerene/semiconductor photocatalysts [4,44]. Typically, the semiconductor is added into the fullerene solution to form adequately dispersed suspensions, and then the newly generating fullerene/semiconductor nanocomposite is obtained through evaporating the solvent. Many semiconductor/fullerene nanocomposites have been synthesized through this method, such as TiO_2_/C_60_, ZnO/C_60_, Bi-based oxides/C_60_ and C_3_N_4_/C_60_ [45,46,47]. Note that the simple adsorption method differs from direct mechanical mixture, because no fullerene is leached out when the nanocomposite is added into the aforementioned organic solvent.

### 3.2. Hydrothermal Synthesis Method

Hydrothermal synthesis is an appealing method for preparing semiconductor/fullerene nanomaterials. In the synthetic procedure, pretreatment of fullerene is imperative and acid-treatment is extensively applied to produce several oxygen-containing active sites on the surface of fullerene. For instance, nitric acid is a common agent to oxide fullerene under reflux condition, further facilitating more efficient combination of fullerene with other semiconductors through hydrothermal procedure. Yu, et al. [48] constructed TiO_2_/C_60_ nanocomposite using this acid-treated C_60_ via a hydrothermal method. Activated C_60_ and Ti(OC_4_H_9_)_4_ (titanium source) were mixed into ethanol/water solution (1:2 v/v) and then the reaction mixture solution was transferred into stainless steel autoclave at 180 °C for 10 h. Similarly, TiO_2_/C_70_ was fabricated by a hydrothermal procedure as well [49]. Very recently, several semiconductor/fullerene photocatalysts were obtained by the hydrothermal method, such as CdS/C_60_, PbMoO_4_/C_60_, BiOCl/C_70_ and C_3_N_4_/(C_60_, C_70_) [50,51]. For example, Cai, Hu, Zhang, Li and Shen [43] constructed CdS/C_60_ photocatalyst via a facile one-pot hydrothermal method, wherein a mixture of acid-treated C_60_, Cd(CH_3_COO)_2_·2H_2_O and _L_-cysteine in water was heated at 200 °C for 10 h in autoclave. Moreover, Ma, Zhong, Li, Wang and Peng [39] synthesized BiOCl/C_70_ photocatalyst via a hydrothermal method and the procedure was presented as follow. Acid-treated C_70_ and Bi(NO_3_)_3_∙5H_2_O were dissolved in glacial acetic acid and KCl solution was slowly added into, and then the mixture solution was transferred into stainless autoclave maintaining at 180 °C for 24 h.

In addition to nitric acid, meta-chloroperoxybenzoic acid (MCPBA) is an effectively alternative oxidizing agent to pretreat fullerene before hydrothermal method. Typically, fullerene and MCPBA are dissolved into benzene, followed by heating reflux for hours to activated the surface of fullerene. For instance, CoS/C_60_ nanocomposite was prepared by using the MCPBA-oxidized C_60_ via hydrothermal method [52]. Namely, MCPBA-oxidized C_60_, CoCl_2_ and Na_2_S_2_O_3_ mixture water solution was heated at 150 °C for 12 h in autoclave, and the CoS/C_60_ nanocomposite was obtained through filter. Similarly, other semiconductor/fullerene nanocomposites were prepared via hydrothermal method with MCPBA-oxidized C_60_, including CdSe/C_60_ and WO_3_/C_60_ [36,53].

### 3.3. Ball Milling Method

Ball-milling is a facile and eco-friendly method to structure solid–solid composites, which could generate stronger intermolecular interactions than physical blends. Recently, the hybridized MoS_2_/C_60_ nanocomposite was obtained through a planetary ball-milling machine [54]. Mixture of MoS_2_ and C_60_ powders was transferred into a ball-milling jar together with stainless steel balls under Ar atmosphere. After operation at 500 rmp for 48 h, the reactant was Soxhlet extracted by CS_2_ to remove the unreacted C_60_. The strong van der Waals (vdW) interactions contributed to formation of MoS_2_/C_60_ heterostructure rather than a covalent conformation, resulting in elevated photocatalytic activity of pure MoS_2_. In addition, a g-C_3_N_4_/C_60_ nanocomposite was fabricated via a ball-milling route as well [55]. Additional LiOH was needed as a catalyst before ball-milling process and the detailed synthetic process of the g-C_3_N_4_/C_60_ was presented in Figure 1. It was firstly proposed in fullerene chemistry that the covalent bonding forms via a four-membered ring of azetidine between C_60_ and g-C_3_N_4_ nanosheets. The LiOH as catalyst breaks π-π carbon bonds of C_60_ to produce C_60_ radicals, then ball-milling activates g-C_3_N_3_ and results in the covalent reaction between g-C_3_N_4_ and C_60_ (Figure 1).

### 3.4. Other Techniques

Sol-gel approach has been used to prepare fullerene-TiO_2_-semiconductor ternary photocatalysts. Meng, et al. [56] fabricated CdS-C_60_/TiO_2_ photocatalyst by a sol-gel method. Firstly, NaS_2_ solution was dropwise added into oxidized C_60_ and (CH_3_COO)_2_Cd·2H_2_O mixed ethanol solution and the collected solids were calcinated at 300 °C to obtain CdS-C_60_ particles. Then, CdS-C_60_ particles were added into titanium (IV) n-butoxide (TNB) solution with constant stirring and CdS-C_60_/TiO_2_ gels were produced in mixed solution under reflux at 70 °C. Finally, the CdS-C_60_/TiO_2_ nanoparticles were obtained after heat treatment at 400 °C. Bai, Wang, Wang, Yao and Zhu [35] proposed a facile thermal treatment method for fabricating g-C_3_N_4_/C_60_ photocatalyst. The procedure was presented as follow: ball-milled C_60_ and dicyandiamide mixture was transferred into a muffle furnace and held at 550 °C for 4 h. Moreover, Li and Ko [57] successfully prepared MoS_2_/C_60_ nanocomposite by a facile heating treatment procedure. The wetness impregnation method is also an effective method for building semiconductor/fullerene photocatalysts. Apostolopoulou, et al. [58] prepared TiO_2_/C_60_ nanoparticles using 1,2-dichloro-benzene as a solvent via a successive incipient wetness impregnation followed by heating at 180 °C. Similarly, a polyhydroxyfullerene (PHF)/titanium dioxide nanotube was prepared by incipient wetness impregnation [59]. Firstly, fullerene was functionalized by NaOH and H_2_O_2_ to obtain PHF (or called fullerenol). Then, PHF was added to TiO_2_ nanotube solution under a wetness impregnation followed by heating at 400 °C. Hence, this provides a new route for coupling fullerenol with other semiconductors to obtain effective photocatalysts.

## 4. The Photocatalytic Application of Fullerene/Semiconductor Photocatalysts

The fullerene/semiconductor photocatalysts have been extensively used for photocatalytic wastewater treatment (pollutant degradation, Cr (VI) reduction, disinfection etc.) and water splitting for H_2_ generation [49,60,61]. Among them, photocatalytic degradation of organic pollutant is ascribed to decomposition of organic molecule structure and photocatalytic disinfection depends on inactivation of microorganisms. However, photocatalytic Cr (VI) reduction focuses on the transformation from Cr (VI) to Cr (III). It is generally believed that chromium (Cr) is among the sixteen most toxic contaminants due to its carcinogenic and teratogenic effect on human. Hence, the World Health Organization (WHO) and the United State Environmental Protection Agency (USEPA) have set the maximum permissible concentration of Cr in drinking water at 0.05 mg/L and 0.1 mg/L [62]. Note that the reduced Cr (III) is far less toxic and more stable than Cr (VI), so the photocatalytic Cr (VI) reduction is a promising method to reduce the chromium toxicity in water.

Furthermore, Table 1 and Table 2 summarize the photocatalytic efficiency in pollutant degradation and H_2_ generation over all kinds of fullerene/semiconductor photocatalysts, respectively. Next, detailed photocatalytic activity and mechanisms will be discussed for various types of semiconductor/fullerene photocatalysts, which is accompanied with analyses of electron transfer routes, free radical reactions and stability of photocatalysts.

### 4.1. Fullerene Based TiO_2_ Photocatalysts

TiO_2_ is the most extensively used photocatalyst due to its easy availability, strong oxidizing ability, and superior photoelectronic properties [63,64]. With a wide band gap (~3.2 eV), TiO_2_ could only be excited under UV light, which limits efficient utilization of solar light [65]. Meanwhile, the fast recombination of photoinduced hole-electron pairs restricts the photocatalytic efficiency of TiO_2_ [66]. It has been proven that coupling fullerene with TiO_2_ is a helpful way to boost the photocatalytic efficiency of pure TiO_2_ both under UV light and visible light irradiation.

Oh, et al. [67] prepared TiO_2_/C_60_ photocatalyst by a heat treatment method with 700 °C. It was shown that the TiO_2_/C_60_ exhibited a more significant effect towards MB degradation with an increase of −ln (C/C_0_) values than that of the original TiO_2_ under UV light illumination. Yu, Ma, Liu and Cheng [48] successfully fabricated mesoporous TiO_2_/C_60_ powders via a hydrothermal method, which demonstrated that C_60_ molecules could be dispersed as monolayer or few layers onto bimodal mesoporous TiO_2_ via covalent bonding. The 0.5 wt % TiO_2_/C_60_ exhibited the best photocatalytic efficiency for acetone decomposition under UV light irradiation and the degradation rate constant (k) was 13.9 × 10^−3^, reaching 3.3-times that of the pure TiO_2_. In this UV-light-driven photocatalytic system, the dominant role of C_60_ is an inhibitor of rapid recombination of photogenerated hole-electron pairs, leading to boost the quantum efficiency of TiO_2_. With respect to TiO_2_/C_60_ photocatalyst, the excited electrons will transfer from TiO_2_ to C_60_ because the conduction band potential of TiO_2_ (−0.5 V vs. NHE) is more negative than that of C_60_/C_60_^•−^ (−0.2 V vs. NHE). Under UV light irradiation, the photogenerated electrons are excited from the VB of TiO_2_ to the CB, leaving holes in the VB. Generally, these holes and electrons incline to fast recombination and only partial carriers take part in redox. However, after C_60_ is tightly coupled with TiO_2_, excited electrons could further transfer to C_60_ due to its excellent electron adsorption capacity, which effectively inhibits recombination of photoinduced carriers and supplies more carriers participating in photocatalytic reaction. Besides, C_60_ derivative (C_60_(CHCOOH)_2_) modified TiO_2_ nanoparticles fabricated by Mu, Long, Kang and Mu [61] showed superior photocatalytic efficiency in Cr (VI) reduction under UV light illumination. Compared with pristine TiO_2_, the C_60_-derivative-modified TiO_2_ nanocomposites exhibited a higher photocatalytic efficiency of 97% for Cr(VI) reduction within 1.5 h UV irradiation. The electron transfer and radical formation procedure is presented as follows, in Equations (1)−(3):(1)$${\mathrm{TiO}}_{2}/C_{60}\overset{h\;v}{\to }C_{60}\left(e^{-}\right)/{\mathrm{TiO}}_{2}\left(h^{+}\right)$$
(2)$${\mathrm{TiO}}_{2}\;\left(h^{+}\right)+{\mathrm{OH}}^{-}\to {\mathrm{TiO}}_{2}+{}^{•}\mathrm{OH}$$
(3)$$C_{60}\left(e^{-}\right)+O_{2}\to C_{60}+{}^{•}O_{2}^{-}$$

In addition to UV-light-driven photocatalytic activity, TiO_2_/C_60_ nanocomposites also exhibit superior photocatalytic capacity under visible light irradiation. In this visible-light-driven photocatalytic system, the introduced C_60_ could typically enhance the photocatalytic activity in two ways at the same time: one is to increase the visible light adsorption, the other is to prolong the lifetimes of photoinduced carriers for participating redox reaction. For example, an investigation was conducted into the visible-light-induce photocatalytic activity of TiO_2_/C_60_ towards MB degradation [88]. In this study, two crystals of TiO_2_ (anatase and rutile) were coupled with C_60_ to assemble photocatalysts and the rutile-C_60_ exhibited significantly superior efficiency than pristine rutile under visible illumination. Grandcolas, et al. [89] synthesized C_60_ modified TiO_2_ nanotubes via a simple impregnation method using ethanol and toluene as co-solvents, and the as-prepared sample exhibited superior efficiency in photocatalytic isopropanol degradation under visible light irradiation. More recently, a polycarboxylic acid functionalized fullerene (C_60_-(COOH)_n_) was coupled with TiO_2_ to obtain a novel photocatalyst TiO_2_/C_60_ nanocomposite via ultrasonication-evaporation method for the first time [68]. For the as-prepared photocatalysts, the introduction of C_60_ obviously decreased the aggregation of pure TiO_2_ nanocomposites (Figure 2a,b), and the C_60_ particles were well-dispersed and closely contacted onto the surface of TiO_2_ (Figure 2c). Compared with pure TiO_2_, the 1 wt % TiO_2_/C_60_ exhibited stronger both UV and visible light absorption, resulting in improving the utilization of light energy (Figure 2d). In order to trace oxidative species involved in the photocatalytic reaction, in situ radical trapping experiments were made, wherein EDTA was used for trapping holes and 1,4-benzoquinone (BQ) was a scavenger for •O_2_^−^. In the presence of EDTA, the photocatalytic degradation efficiency to RhB was dramatically retarded, and a similar trend was also observed with BQ addition (Figure 2e). These results meant that the photoinduced h^+^ and •O_2_^−^ were involved in the photocatalytic reaction. Under visible light illumination for 150 min, 1 wt % TiO_2_/C_60_ nanocomposite showed 95% degradation efficiency to RhB, which was significantly higher than pristine TiO_2_ (Figure 2f). To further check the stability of the TiO_2_/C_60_ photocatalyst, the recovered composite was used for repeatedly photocatalytic degradation experiment towards RhB. After five repeated experiments under visible light irradiation for 150 min, the degradation efficiency decreased from 95% to around 80%. There is no doubt that the long-term stability of photocatalysts is particularly vital to practical application. Therefore, future work could focus on the synthesis method of TiO_2_ to improve the stability of TiO_2_/C_60_ photocatalysts. For example, Bastakoti, et al. [90] reported a high-efficiency method to fabricate more stable mesoporous metal oxides (including TiO_2_, Ta_2_O_5_ and Nb_2_O_5_).

Compared to TiO_2_, C_70_ is a close-shell configuration consisting of 35 bonding molecular orbitals with 70 *p*-electrons [91]. Similar to C_60_, C_70_ has higher electron acceptability and higher efficiency of light harvesting over TiO_2_ [92,93]. Thus, C_70_ is a promising alternative to boost the photocatalytic efficiency of TiO_2_. Cho, et al. [94] fabricated both TiO_2_/C_60_ and TiO_2_/C_70_ nanowire to estimate their photocatalytic activity. In this study, TiO_2_/C_70_ showed a significantly stronger absorbance within 400~650 nm and a lower photoluminescence spectra (PL) than TiO_2_/C_60_. This means the C_70_ displayed better efficiency in boosting visible light absorption and inhibiting recombination of hole-electron pairs than C_60_. Accordingly, the TiO_2_/C_70_ nanowire displayed higher photocatalytic activity for MB degradation than TiO_2_/C_60_ in the visible light irradiation. Furthermore, Wang, Liu, Dai, Cai, Chen, Yang and Huang [69] assembled TiO_2_-C_70_ hybrids using acid-treated C_70_, Ti(SO_4_)_2_ and cetyltrimethylammonium bromide (CTAB) by a hydrothermal method. It was proven that the 8.5 wt % TiO_2_-C_70_ showed the best photodegradation efficiency to sulfathiazole under visible light, which was 4.2 times that of TiO_2_ + C_70_ mixture and 1.6 times that of the corresponding TiO_2_-C_60_ nanocomposite, respectively. The SEM and TEM images of 18 wt % TiO_2_-C_70_ nanocomposite are shown in Figure 3a,b. After C_70_ introduction, the surface of TiO_2_-C_70_ nanocomposites were uneven, which could increase the BET of the as-prepared samples. The C_70_ particles were well-dispersed onto the outer boundary of TiO_2_ composites, and it was estimated that a monolayer of C_70_ was covered onto the surface of TiO_2_. Additionally, the TiO_2_-C_70_ exhibited better light absorption and higher separation efficiency of hole-electron pairs than those of TiO_2_-C_60_ and pure TiO_2_ (Figure 3c,d). It is important to highlight that novel mechanisms of TiO_2_/fullerene were proposed for photocatalytic pollutant degradation [69]. From two aspects, UV light and visible light irradiation, the mechanisms are described as follows. Under UV light illumination, electrons are excited from VB to CB of TiO_2_, leaving holes in the VB. Then the CB electrons of TiO_2_ could rapidly transfer to C_70_, because the CB potential of TiO_2_ (0.5 V vs. NHE) is more negative relative to C_70_/C_70_^−^ (0.2 V vs. NHE). In the meantime, the ground-state C_70_ is excited to a transient-state ^1^C_70_*, then undergoes rapid intersystem crossing (ISC) to a lower lying triplet state ^3^C_70_*. In this system, excited electrons can be injected into the three-states transform procedure of C_70_, resulting in suppression of their falling back to the VB of TiO_2_. Hence, this process effectively inhibits the recombination of photoinduced hole-electron pairs, so as to elevate the photocatalytic activity of TiO_2_/C_70_ nanocomposite (Figure 3e). On the other hand, a viewpoint of mid-gap band was proposed for TiO_2_/C_70_ with respect to the visible-light-driven photocatalytic mechanism. It was pointed out that the mid-gap band came into being between TiO_2_ and C_70_ ascribing to the strong chemical boning of these two materials. The electron transfer route was obviously distinct from those TiO_2_/fullerene photocatalysts in previous studies. Namely, visible light excites electrons from the VB of TiO_2_ to the mid-gap band and then from the mid-band to the CB of TiO_2_, leaving holes in the VB (Figure 3e). This procedure significantly prolongs the lifetime of the photoinduced carriers and facilitates the separation of hole-electron pairs for participating in photocatalytic reaction. Accordingly, the effectively separated holes and electrons participate in generation of reactive radical species. The e^⁻^ could react with dissolved O_2_ to produce •O_2_^−^ and h^+^ react with H_2_O to produce •OH, then these radical species cause the degradation of sulfathiazole. To research the stability of TiO_2_/C_70_ nanocomposite, the recycled sample was dried for subsequently repeated experiments. After 5 runs (totally 15 h visible light illumination), the degradation efficiency to sulfathiazole over TiO_2_/C_70_ slightly declined, remaining over 90%. This evidenced that the aforementioned hydrothermal method was effective in building stable TiO_2_/fullerene photocatalysts.

Oh and Ko [95] fabricated Pt-fullerene/TiO_2_ nanocomposites via in-situ growth method using Pt-treated oxidized fullerene and TNB. Firstly, fullerene was oxidized by MCPBA and treated through ion exchange using potassium hexachloroplatinate (IV) (K_2_[PtCl_6_]), wherein Pt-treated oxidized fullerene was obtained. Then, Pt-fullerene was added into TNB solution (titanium source) for fabricating Pt-fullerene/TiO_2_ nanocomposite via a sol-gel method under mild condition (50 °C). The as-prepared sample exhibited elevated performance under UV light and the order of photocatalytic efficiency for MB degradation was: Pt-fullerene/TiO_2_ > fullerene/TiO_2_ ˃ pristine TiO_2_, due to the synergetic effects of Pt, oxidized-fullerene and TiO_2_. In this study, it was proposed that Pt-fullerene was homogeneously covered with TiO_2_ particles, wherein TiO_2_ would be mounted in a 3-dimensional matrix. It was concluded that three factors contributed to the superior photocatalytic activity of Pt-fullerene/TiO_2_, including photocatalytic reaction of the supported TiO_2_, decomposition of the organo-metallic reaction by the Pt compound and energy transfer effects of fullerene. Through the same method, a number of metal-treated fullerene/TiO_2_ composites were prepared for photocatalytic application as well, such as Fe-C_60_/TiO_2_, V-C_60_/TiO_2_ and Pd-C_60_/TiO_2_ [96,97]. For instance, Meng, Zhang, Zhu, Park, Ghosh, Choi and Oh [76] fabricated M-fullerene/TiO_2_ (M representing Pt, Y or Pd) composites to compare their photocatalytic efficiency. Among these samples, the Pd-fullerene/TiO_2_ showed the best photocatalytic activity for MB decomposition under UV light, due to its better dispersion and larger BET surface over the Pt-fullerene/TiO_2_ and Y-fullerene/TiO_2_. Further results indicated that the synergistic effects between Pd and fullerene improves the photocatalytic activity of TiO_2_, including enhancement of light adsorption by fullerene and Pd as the final electron-acceptor. More recently, Islam, Hangkun, Ting, Zubia, Goos, Bernal, Botez, Narayan, Chan and Noveron [77] prepared AuNPs-TiO_2_-C_60_ composites, wherein the introduction of C_60_ significantly boosts the photoactivity and photostability of AuNPs-TiO_2_. It was reported that C_60_ played threefold roles in the preparation of AuNPs-TiO_2_-C_60_. The introduction of C_60_ decreased the size of AuNPs (5 nm) and effectively prevented its agglomeration on the surface of TiO_2_, as well as linked AuNPs to TiO_2_ surface without any functionalization. The AuNPs-TiO_2_-C_60_ had a broader light adsorption region over pristine TiO_2_ and AuNPs-TiO_2_ nanocomposites, ranging from 500 to 650 nm. The 4.76% optimal AuNPs-TiO_2_-C_60_ sample showed 95% photodegradation efficiency towards MO after visible light irradiation for 160 min, which was 2 times higher than pristine TiO_2_.

Meng and his co-workers assembled a series of semiconductor/fullerene/TiO_2_ ternary photocatalysts via sol-gel method, such as TiO_2_/CdS/C_60_, TiO_2_/CdSe/C_60_ and TiO_2_/WO_3_/C_60_ nanocomposite [56,78,98]. For example, the TiO_2_/CdS/C_60_ photocatalyst exhibited superior efficiency in photocatalytic pollution degradation and the MO degradation rate (K) of these as-formed nanocomposites was in an order: TiO_2_/CdS/C_60_ > TiO_2_/C_60_ > TiO_2_ > TiO_2_/CdS. In addition, Lian, Xu, Wang, Zhang, Xiao, Li and Li [84] successfully fabricated C_60_ decorated TiO_2_/CdS mesoporous photocatalyst via an evaporation combined with ion-exchanged method. It is noteworthy that the BET of the TiO_2_/CdS/C_60_ photocatalyst was actually lower than that of the TiO_2_/CdS composite, which resulted from the fact that the C_60_ was inset into the pore of this mesoporous composite. Compared with CdS/TiO_2_, the TiO_2_/CdS/C_60_ presented stronger visible light adsorption, lower recombination of photogenerated hole-electron pairs and higher photocurrent density, thus resulting in highly effectively photocatalytic ability for H_2_ production (Figure 4a–c). In Na_2_S-Na_2_SO_3_ reaction solution, the H_2_ generation rate of the optimal 0.5 wt % TiO_2_/CdS/C_60_ photocatalyst was 6.03µmol h^−^^1^ g^−1^ with 2.0% quantum efficiency (QE) under visible light, which was obviously higher than the rate of TiO_2_/CdS (0.71 µmol h^−1^ g^−1^). As concluded in this study, C_60_ could enhance the light adsorption and facilitate the separation of photogenerated hole-electron pairs, as well as serve as H_2_ generation site for adsorbing and reducing H⁺ ions. The electron transfer route and reaction mechanism are presented in Figure 4d. Under visible light illumination, electrons in the VB of CdS are excited into the CB firstly. Then the excited electrons in the CB of CdS rapidly transfer into the CB of TiO_2_, because the conduction band potential of the former is more negative than that of the later. Finally, the electrons in the CB of TiO_2_ transfer to C_60_, which provide reaction sites for reducing H⁺ to H_2_. Meanwhile, the leaving holes in the VB of the CdS are consumed by S^2−^ and SO_3_
^2−^ to facilitate H_2_ generation efficiency. Moreover, Chai, Peng, Zhang, Mao, Li and Zhang [85] developed a TiO_2_-C_60_-dCNTs photocatalyst, which was reported to enhance the photocatalytic H_2_ production under UV light illumination. At a 5 wt % loading amount of C_60_, it exhibited the highest H_2_ production rate of 651 µmol h^−1^, which is 2.9 times that of bare TiO_2_.

Fullerenol (C_60_(OH)_x_), also called polyhydroxyfullerene (PHF), is a water-soluble fullerene derivative [99,100]. Typically, PHF could be prepared using fullerene via acid hydrolysis or alkali hydrolysis method [101,102,103]. In the earlier time, Krishna and his co-workers found that the addition of PHF in solution could elevate the photocatalytic activity of TiO_2_ under UV light illumination [104]. The reaction solution with PHF + TiO_2_ showed 2.6 times faster of photocatalytic organic dye degradation and 1.9 times faster of *Escherichia coli* inactivation than that of solution with TiO_2_ alone. While the hydroxylated fullerene (PHF) changes the electronic properties and decreases the electron affinity of fullerene, further studies were conducted by his group to explore the mechanisms of PHF to enhance the photocatalytic activity of TiO_2_ [105]. It was proposed that PHF covers onto the surface of TiO_2_ by electrostatic interactions in solution. The electron paramagnetic resonance (EPR) results showed that higher production rate of •OH was achieved under UV light after addition of PHF in solution, which contributed to enhancement of TiO_2_ photocatalytic activity. However, PHF alone in solution did not generate •OH under UV light, which suggested that synergistic effects come into being between PHF and TiO_2_. A hypothesis was proposed that PHF can scavenge the photo-generated electrons from TiO_2_ and meanwhile the synergistic effects of PHF and TiO_2_ can induce more •OH generation for enhancing the photodegradation activity. Furthermore, Park, et al. [106] proposed a new approach of CT in PHF/TiO_2_ system under visible light named as “surface-complex CT” procedure, which had not yet been proposed in TiO_2_/C_60_ system before. Note that fullerol has numerous hydroxyl groups which may link to the surface of TiO_2_ through the CT-complex route (Equation (4)). The photocurrents (I_ph_) of the PHF/TiO_2_-coated electrodes were examined, confirming that the transfer orientation of photogenerated electrons was from PHF to TiO_2_. In such surface-complex CT procedure, PHF serves as a photosensitizer in which electrons could be excited into CB of TiO_2_ under visible light irradiation, hence elevating the photocatalytic efficiency of TiO_2_. The reaction mechanism of radical generation is presented in Equations (5)−(8). Similar CT situations have been proposed in benzene/TiO_2_ system and even polycyclic aromatic hydrocarbons (PAH) physical-adsorbed TiO_2_ system as well [107,108].
(4)$$C_{60}{\left(\mathrm{OH}\right)}_{x}+\equiv \mathrm{Ti}–\mathrm{OH}\to \equiv \mathrm{Ti}–O–C_{60}{\left(\mathrm{OH}\right)}_{x-1}+H_{2}O$$
(5)$$\mathrm{Fullerol}/{\mathrm{TiO}}_{2}+hv(\lambda >420\;\mathrm{nm})\to \left({\mathrm{fullerol}}^{•+}\right)/{\mathrm{TiO}}_{2}\left({e_{\mathrm{cb}}}^{-}\right)$$
(6)$$\left({\mathrm{fullerol}}^{•+}\right)/{\mathrm{TiO}}_{2}\left({e_{\mathrm{cb}}}^{-}\right)\to \mathrm{fullerol}/{\mathrm{TiO}}_{2}$$
(7)$$\mathrm{Fullerol}/{\mathrm{TiO}}_{2}\left({e_{\mathrm{cb}}}^{-}\right)\to {\mathrm{TiO}}_{2}/\left({\mathrm{fullerol}}^{•-}\right)$$
(8)$$\left({\mathrm{fullerol}}^{•+}\right)/{\mathrm{TiO}}_{2}+0.5H_{2}O\to \mathrm{fullerol}/{\mathrm{TiO}}_{2}+H^{+}+0.25O_{2}$$

Bai, Krishna, Wang, Moudgil and Koopman [80] assembled a PHF/TiO_2_ nanocomposite by a physically mixing method in aqueous suspension and then coated the as-prepared sample onto grout substrate to examine its photocatalytic activity. The nanocomposite coating at a TiO_2_/PHF ratio of 0.01 exhibited the best photocatalytic efficiency under UV light irradiation. Accordingly, the 0.01 TiO_2_/PHF photocatalyst coating exhibited 2 times higher of Procion red MX-5B photocatalytic degradation efficiency and 3 times higher of photocatalytic spores of *Aspergillus niger* inactivation than these of bare TiO_2_ coating. Similarly, Hamandi, Berhault, Dappozze, Guillard and Kochkar [59] fabricated TiO_2_/PHF nanotubes using PHF and TiO_2_ nanotubes via wetness impregnation together with heat treated at 400 °C Under UV light illumination, the optimum 1% TiO_2_/PHF nanotubes showed a rate constant values (K_exp_) of 94.7 µmolL^−1^ min^−^^1^ for photocatalytic degradation towards formic acid, while TiO_2_ nanotubes alone exhibited the K_exp_ of 72.6 µmolL^−1^ min^−1^. Moreover, Lim, Monllor-Satoca, Jang, Lee and Choi [81] developed a Nb-TiO_2_/fullerol nanocomposite, which proved an elevated visible-light-driven photocatalytic performance. In brief, Nb-doped TiO_2_ was fabricated by a sol-gel method then the Nb-TiO_2_ was dispersed into fullerol solution buffered at pH 3 with HClO_4_. After stirring for 3 h, the filtered solids were dried at 80 °C to acquire brownish particles designated as Nb-TiO_2_/fullerol. The Nb-TiO_2_/fullerol showed more effectively photocatalytic performance for the reduction of Cr (VI), oxidation of iodide and degradation of 4-chlorophenol than naked TiO_2_, Nb−TiO_2_ and TiO_2_/fullerol under visible light. These results indicated that the synergistic effects between fullerol and Nb improved the photocatalytic activity of TiO_2_. It was proved that Nb doping induced vacancies of TiO_2_ by ionic substitution of Nb^5+^ in Ti^4+^ position, which could suppress the photoinduced hole-electron pairs recombination by trapping electrons. Notably, the fullerol significantly enhanced the visible light absorption of Nb-TiO_2_ through a surface-complex CT mechanism. Under visible light irradiation, electrons will be excited from HOMO of fullerol to CB of TiO_2_ and then from CB to vacancies, which effectively enhances the charge transport and prolongs the lifetime of photoinduced carriers. Another advantage was proposed that the Nb-TiO_2_/fullerol showed more highly photochemical stability over typical dye-sensitized-TiO_2_.

### 4.2. Metal Oxides (Except TiO_2_)/Fullerene Photocatalyst

In addition to TiO_2_, other metal oxides have also been promising materials in photocatalytic application [109,110,111]. Fullerene (C_60_ and C_70_) has some advantages to enhance the photocatalytic efficiency of these metal oxides, such as enhancing the light absorption and inhibiting recombination of photogenerated hole-electron pairs. Accordingly, a number of metal-oxide/fullerene photocatalysts have been successfully synthesized and extensively applied in photocatalytic pollutant degradation and H_2_ evolution via water splitting, such as ZnO/C_60_ (C_70_), WO_3_/C_60_ and SnO_2_/C_60_ [25,112].

Similar to TiO_2_, ZnO is an alternative photocatalyst with a band gap of 3.3 eV [113]. Generally, the absorption edge of pristine ZnO locates the near UV region, which usually restricts its photocatalytic efficiency. Meanwhile, the susceptibility to photocorrosion is also another barrier of ZnO for satisfactory photocatalytic performance. Fu, Xu, Zhu and Zhu [70] successfully prepared a C_60_ hybridized ZnO nanocomposite by a simple absorption method, and the sample with 1.5 wt % C_60_ exhibited the best photocatalytic sufficiency in MB degradation. The 1.5 wt % ZnO/C_60_ composite showed 95% photocatalytic degradation sufficiency towards MB under UV light, which was 3-times as high as that of bare ZnO. In such system, the elevated performance was ascribed to the improved light adsorption and a higher separation efficiency of photoinduced hole-electron pairs. During the photocatalysis process, the photogeneration holes could easily react with surface oxygen atom, leading to fast decline of photocatalytic activity of ZnO. When C_60_ was covered on ZnO, the activity of surface oxygen atoms was effectively reduced so that more holes could participate in photocatalytic reaction. Furthermore, the photocorrosion experiment indicated that the C_60_-hybridized ZnO nanocomposite did not show obviously decline of photocatalytic sufficiency even after illumination under UV light for 50 h, which was highly superior in stabilization than bare ZnO. Hence, the introduction of C_60_ effectually suppress photocorrosion of ZnO. Similarly, Hong, et al. [114] successfully prepared a ZnO/C_70_ nanocomposite via a heat treatment method, which exhibited superior photocatalytic degradation of organic dyes.

Recently, Tahir, Nabi, Rafique and Khalid [53] proposed the elevated photocatalytic efficiency of WO_3_/C_60_ nanocomposite for dye degradation and H_2_ evolution. The optimized 4 wt % WO_3_/C_60_ sample showed the best photodegradation ability and the degradation efficiency in MB, RhB and MO under visible light illumination was 93%, 92% and 91%, respectively. Meanwhile, the H_2_ evolution rate of the 4 wt % WO_3_/C_60_ was 2-times higher than that of bare WO_3_. After coupling with C_60_, the band gap of WO_3_/C_60_ nanocomposites were lower than that of bare WO_3_, which could excite more electrons of semiconductor WO_3_ from VB to CB. The BET surface area of these composites was also significantly increased, which could enhance the adsorption reaction. Based on the aforementioned study, Shahzad, Tahir and Sagir [87] constructed a novel heterogeneous photocatalyst WO_3_/fullerene@Ni_3_B/Ni(OH)_2_ for H_2_ production. As a co-catalyst, Ni_3_B/Ni(OH)_2_ was loaded onto WO_3_/fullerene thin film by a facile photo-deposition technique. The optimal WO_3_/fullerene@1.5%Ni_3_B/Ni(OH)_2_ presented an outstanding photocatalytic efficiency in H_2_ generation, reaching 1578µmol h^−1^ g^−^^1^. In this system, it was proposed that three factors mainly contributed to the superior performance, including inhibition for recombination of photogeneration hole-electron pairs, more active sites for photocatalytic reaction and synergistic effect between nanocomposite and co-catalyst. Even earlier, a ternary photocatalyst WO_3_/C_60_/TiO_2_ was successfully prepared via a sol-gel method. Its photocatalytic performance in MO degradation was higher than that of WO_3_/C_60_ or TiO_2_/C_60_, which means these three materials synergistically enhance the photocatalytic activity [78].

In addition, Song, Zhang, Zeng, Wang, Ali and Zeng [83] fabricated a series C_60_ modified Fe_2_O_3_ polymorphs (α-, γ- and β-Fe_2_O_3_) photocatalysts via a simple adsorption method. These as-prepared samples showed superior photocatalytic efficiency in H_2_ production and even extremely outstanding effects were observed in the presence of fluorescein. Under visible light irradiation, the photocatalytic capacity was in the order: 1C_60_/β-Fe_2_O_3_ > 1C_60_/γ-Fe_2_O_3_> γ-Fe_2_O_3_ > 1C_60_/α-Fe_2_O_3_> β-Fe_2_O_3_ > g-C_3_N_4_ > α-Fe_2_O_3_. Behera, Mansingh, Das and Parida [71] proposed a ZnFe_2_O_4_-fullerene photocatalyst for norfloxacin decomposition and Cr (VI) reduction, wherein fullerene introducing significantly improved the photocatalytic capacity of ZnFe_2_O_4_. Moreover, Song, Huo, Liao, Zeng, Qin and Zeng [82] successfully prepared a novel photocatalyst Cr_2-x_Fe_x_O_3_/C_60_ via a simple adsorption method. In this study, α-Fe_2_O_3_ (~2.2eV) and Cr_2_O_3_ (~3.4eV) were integrated through a sol-gel method in order to construct Cr_2-x_Fe_x_O_3_ with a suitable band gap for H_2_ generation. While poor electron conducting ability limited its further application, the introducing of C_60_ was an effective way. The optimal 3%C_60_/Cr_1.3_Fe_0.7_O_3_ sample presented the H_2_ generation rate of 220.5 µmol h^−1^ g^−1^, which was about 2-times of the bare Cr_1.3_Fe_0.7_O_3_ composite.

### 4.3. Metal Sulfide/Fullerene Nanocomposites

Nowadays, metal sulfide semiconductors have attracted extensive attentions in photocatalytic application due to their distinctive optical-electrical characteristic [115,116,117]. CdS is an appealing photocatalyst with narrow bandgap (2.2~2.4 eV) exhibiting superior visible-light respond [118]. While fast recombination of hole-electron pairs and photocorrosion effect are the two main obstacles of naked CdS, which inhibits its photocatalytic efficiency. Coupling with fullerene was proved to be an effective way to boost the photocatalytic performance of CdS. Accordingly, Cai, Hu, Zhang, Li and Shen [43] successfully assembled a CdS/C_60_ nanocomposite via one-pot hydrothermal synthesis. The as-prepared samples showed better separation efficiency of photoinduced hole-electron pairs and higher photocurrent density than pure CdS (Figure 5a,b). Thus, the improvement of the aforementioned features contributed to a highly photocatalytic activity over CdS/C_60_ nanocomposite. The optimal H_2_ production rate of 0.4 wt % CdS/C_60_ was 1.73 mmol h^−1^ g^−1^ under visible light illumination, which was 2.3-times higher than that of naked CdS (Figure 5c). Its photocatalytic degradation efficiency towards RhB achieved 97% in 40 min (Figure 5d). Furthermore, the photostability of CdS was significantly boosted after CdS combining with C_60_, and 97.8% of RhB degradation efficiency actually retained after three recycles (Figure 5e). In order to estimate the stability of CdS/C_60_, the released Cd^2+^ concentration was determined in remaining solution after three cycles for RhB degradation (Figure 5f). The Cd^2+^ concentration was 381.3µg/L in solution with naked CdS while it was 51.9µg/L in solution with 0.4 wt %CdS/C_60_ nanomaterial, in which the former was 7.3 times of the later. The results above indicated that C_60_ could effectively inhibit the photocorrosion and boost the stability of CdS. Furthermore, Meng, Peng, Zhu, Oh and Zhang [56] assembled a novel ternary CdS/TiO_2_/C_60_ photocatalyst via a sol-gel method. The introduction of C_60_ definitely induced 56% increasement of the BET surface of the CdS/TiO_2_ composite, which could enhance the adsorption effect. Under the same condition, the MO degradation rate (K) of these nanocomposites was in an order: CdS/TiO_2_/C_60_ > TiO_2_/C_60_ > TiO_2_ > CdS/TiO_2_. This meant the CdS/TiO_2_/C_60_ composite obtained superior photocatalytic capacity owing to the synergistic reaction of C_60_, TiO_2_ and CdS. It was concluded in this study that the synergistic effects were as follows: (1) C_60_ could increase the quantum efficiency and charge transfer, as well as enhance the adsorption effect of the ternary photocatalyst; (2) Combining CdS with TiO_2_ endows the photocatalyst with a suitable bandgap for visible-light respond and a more effective electron transfer route for generating more •OH and •O_2_^−^.

In addition, Meng and co-workers assembled CoS/C_60_ and AgS/C_60_ nanocomposites for pollutant decomposition [52,119]. Superior photocatalytic efficiency was obtained in these photocatalysts after the introduction of C_60_, since C_60_ is an energy sensitizer that could improve the quantum efficiency and boost charge transfer efficiency. More recently, Guan, Wu, Jiang, Zhu, Guan, Lei, Du, Zeng and Yang [54] fabricated a MoS_2_/C_60_ heterostructure photocatalyst via a ball milling method. The method does not need solvent to dissolve MoS_2_ and C_60_ and can significantly increase the BET surface of product. In this study, it was the first time to propose that a van der Waals heterostructure formed between MoS_2_ and C_60_ through ball milling, detailly wherein C_60_ nanoparticles bounded onto the edge of the exfoliated MoS_2_ nanosheet by non-covalent bond. Noteworthily, the CB minimum of MoS_2_/C_60_ was more negative than that of ball-milled MoS_2_ and the VB maximum of the former was more positive than that of the later. Thus, the as-prepared MoS_2_/C_60_ photocatalyst featured more suitable band gap for elevating the H_2_ evolution. Under visible light irradiation, the optimal 2.8 wt % MoS_2_/C_60_ sample exhibited the photocatalytic H_2_ production rate of 6.89 mmol h^−1^ g^−1^ in the presence of EY as a photosensitizer, which was 9.5 times higher than that of ball-milled MoS_2_ without C_60_.

### 4.4. Bismuth-Based Semiconductor/Fullerene Composites

Bismuth-based semiconductors have been proven promising materials for photocatalytic application, including BiOX (X = Br, Cl and I), Bi_2_WO_6_, BiVO_4_, Bi_2_MoO_6_, and so on [2,120,121]. Considerable research efforts have been devoted to couple these bismuth-based semiconductors with fullerene (C_60_ or C_70_) and enhanced photocatalytic performance could be obtained. For example, Zhu, Xu, Fu, Zhao and Zhu [40] successfully prepared C_60_ modified Bi_2_WO_6_ photocatalyst via a simple absorbing process, and 1.25 wt % Bi_2_WO_6_/C_60_ displayed 5.0-times the photocatalytic degradation activity towards MB with respect to unmodified Bi_2_WO_6_. Similarly, Ma, Zhong, Li, Wang and Peng [39] fabricated C_70_ modified BiOCl by an in-situ preparation procedure and superior photocatalytic performance was observed. Under solar irradiation for 30 min, 49.7% of RhB was degraded over pure BiOCl while 99.8% of RhB could be degraded over 1 wt % BiOCl/C_70_. In addition, Li, Jiang, Li, Lian, Xiao, Zhu, Zhang and Li [73] successfully developed Bi_2_TiO_4_F_2_/C_60_ photocatalyst via a solvothermal method, which was a hierarchical microsphere structure. The introduction of C_60_ can increase the photocurrent of the as-prepared sample, resulting from more efficient mobility efficiency of the charge carriers (Figure 6a). Owing to strong combining and heterojunction formation, the Bi_2_TiO_4_F_2_/C_60_ nanocomposite showed obviously elevated photocatalytic capacity for degrading RhB relative to bare Bi_2_TiO_4_F_2_ under visible light irradiation (Figure 6b). Meanwhile, the photocatalyst exhibited excellent stabilization as well and highly photocatalytic efficiency of RhB degradation (≈ 80%) was maintained even after eight circles (Figure 6c). The photocatalytic mechanisms of Bi_2_TiO_4_/C_60_ nanocomposite are described in Figure 6d. Apart from organic pollutant, bromate (BrO^3^^−^) also exhibits biotoxicity to aquatic organisms and human since its properties non-biodegradation and accumulation. A strategy for controlling BrO^3−^ pollution is to reduce it to Br^⁻^ which is naturally present in surface water bodies. Therefore, Zhao, Liu, Shen and Qu [46] studied the photocatalytic performance of Bi_2_MoO_6_/C_60_ for removal BrO^3−^ under visible light. After modification with C_60_, Bi_2_MoO_6_/C_60_ exhibited sharply increase in photocatalytic reduction of BrO^3−^, attributed to the enhanced separation rate of photogenerated electron-hole pairs.

### 4.5. Carbon Nitride/Fullerene Composites

Graphitic carbon nitride (g-C_3_N_4_) is an attractive metal-free photocatalyst, which was developed by Wang et al. in 2009 [122]. The pristine g-C_3_N_4_ features a medium band gap (2.5~2.7 eV) with good visible light response [123]. Currently, this effective organo-photocatalyst has been widely used for pollutant degradation, water splitting and CO_2_ reduction [124,125]. However, pristine g-C_3_N_4_ exhibits insufficient solar absorption and rapid recombination of photogenerated carriers, which limits its photocatalytic efficiency. It has been proven that coupling g-C_3_N_4_ with fullerene is an effective way to enhance the photocatalytic sufficiency in pollutant degradation and H_2_ evolution. Recently, a series of g-C_3_N_4_/fullerene nanocomposites have been fabricated and they showed elevated photocatalytic efficiency.

Chai, Liao, Song and Zhou [45] prepared g-C_3_N_4_/C_60_ nanocomposites via a simple adsorption method. After C_60_ introduction, g-C_3_N_4_/C_60_ nanocomposites enhanced the visible light absorption without changing the absorption edge of g-C_3_N_4_, as well as lowered the recombination of photogenerated hole-electron pairs. The 1 wt % C_3_N_4_/C_60_ showed the highest photodegradation performance towards RhB, which could reach 97% degradation efficiency under visible light after 60 min. The reaction process could be proposed as follows (Equations (9)–(12)):(9)$$C_{60}/C_{3}N_{4}\overset{h\;v}{\to }\;C_{60}\left(e^{-}\right)/C_{3}N_{4}\left(h^{+}\right)$$
(10)$$C_{60}\left(e^{-}\right)+O_{2}\to C_{60}+{}^{•}O_{2}^{-}$$
(11)$${}^{•}{O_{2}}^{-}+2e^{-}+2H^{+}\to {}^{•}\mathrm{OH}+{\mathrm{OH}}^{-}$$
(12)$$\mathrm{RhB}+h^{+}({}^{•}\mathrm{OH},{}^{•}{O_{2}}^{-})\to \mathrm{products}$$

Bai, Wang, Wang, Yao and Zhu [35] introduced C_60_ into g-C_3_N_4_ matric via thermal treatment of C_60_ and dicyandiamide mixture at 550 °C and the obtained nanocomposites exhibited higher photooxidation degradation efficiency towards phenol and MB. Relative to physical blend, this thermal treatment gave rise to strong interface interaction between g-C_3_N_4_ and C_60_. The g-C_3_N_4_/C_60_ nanocomposite exhibited higher specific surface area than pristine g-C_3_N_4_, leading to more active sites for catalytic reaction. The photocurrent value of g-C_3_N_4_/C_60_ is 4.0-times that of g-C_3_N_4_, which was highly responsible for enhancing the photocatalytic activity of pristine g-C_3_N_4_. Moreover, the introduction of C_60_ decreased the band gap of C_3_N_4_, wherein the value of g-C_3_N_4_ and g-C_3_N_4_/C_60_ were severally 2.70 eV and 2.58 eV, respectively. The calculation results showed that the valence band maximum (VBM) of C_3_N_4_/C_60_ is 0.17V lower than that of g-C_3_N_4_, which meant a stronger oxidizing capacity. Under visible light, the degradation ability of g-C_3_N_4_/C_60_ towards phenol and MB were 2.9- and 3.2-times as high as that of pristine g-C_3_N_4_, respectively. In addition, a series of C_3_N_4_/fullerene (C_60_, C_70_) photocatalysts were prepared by Ouyang, et al. [126] via a hydrothermal method for disinfection of bacterial pathogens under visible light irradiation. Regarding disinfection of *E. coli O157:H7*, both C_3_N_4_/C_60_ and C_3_N_4_/C_70_ hybrids showed stronger bacterial inactivation than pristine C_3_N_4_ after 4 h irradiation, and the C_3_N_4_/C_70_ exhibited the best performance. Note that both •O_2_^−^ and •OH were identified as radical species to destruct bacterial cell in the solution under the visible light irradiation.

Coupling g-C_3_N_4_ with C_60_ could elevate the photocatalytic ability to H_2_ generation as well. For instance, Chen, Chen, Guan, Zhen, Sun, Du, Lu and Yang [55] successfully synthesized a covalent bonding g-C_3_N_4_/C_60_ nanocomposite via ball milling with LiOH as catalyst, which was the first time using this method for fabricating semiconductor/C_60_ nanocomposite. As depicted in XRD image, the lattice structure of g-C_3_N_4_ nanomaterial was changed after the attachment of C_60_ component (Figure 7a). It was proven that the covalent bonds were formed in the C_3_N_4_/C_60_ nanocomposite and a new peak at 399.5 eV was detected in XPS spectra, which was ascribed to N-C_60_ bond (Figure 7b). In this study, a new viewpoint was proposed that C_60_ forms covalent bond with g-C_3_N_4_ by a four-membered ring of azetidine. However, g-C_3_N_4_/C_60_ alone hardly exhibited the photocatalytic capacity in H_2_ evolution, thus additional photosensitizer was necessary. Under visible light, the H_2_ generation rate of g-C_3_N_4_/C_60_ was 266 µmol h^−1^g^−1^ using EY as a photosensitizer, which was 4.0 times higher than that of g-C_3_N_4_ in the same condition (Figure 7c). As depicted in Figure 7d, the mechanisms of g-C_3_N_4_/C_60_ nanocomposite are described for photocatalytic H_2_ generation. Recently, a novel g-C_3_N_4_/graphene/C_60_ composite was successfully prepared and significant enhancement for H_2_ evolution ability of the photocatalyst was observed [19]. In the presence of Pt (cocatalyst) and triethanolamine (sacrificial agent), the H_2_ evolution rate of the g-C_3_N_4_/graphene/C_60_ was 5449.5µmol g^−1^ within 10 h, which was 50.4 and 4.24 times that of g-C_3_N_4_/graphene and g-C_3_N_4_/C_60_, respectively. It means that C_60_ and graphene mutually reinforced synergy in H_2_ generation of g-C_3_N_4_, owing to high conductivity of graphene and excellent electron-attracting capacity of C_60_. Meanwhile, the quantum yield of g-C_3_N_4_/graphene/C_60_ reached 7.2% within 72 h.

### 4.6. Other Semiconductor/Fullerene Photocatalysts

Other semiconductors have also been coupled with fullerene to boost the photocatalytic activity. For example, Dai, Yao, Liu, Mohamed, Chen and Huang [50] successfully fabricated a PbMoO_4_-C_60_ photocatalyst via a hydrothermal method. After introduction of C_60_, no obvious change was visible in lattice structure of PbMoO_4_, but defects were observed on the surface of PbMoO_4_ (Figure 8a,b), which could be owed to the decreased crystallinity. As depicted in energy dispersive spectrometry (EDS), a great deal of C element was evenly dispersed on the surface of PbMoO_4_-C_60_ nanocomposite, which was regarded as a layer coating of C_60_ moiety (Figure 8c,d). Upon the attachment of the C_60_ moiety, the PbMoO_4_-C_60_ composite displayed obvious enhancement of both UV and visible light absorption (Figure 8e). Meanwhile, the Eg of 5.0 wt % PbMoO_4_-C_60_ (3.08 eV) was narrower than that of pure PbMoO_4_ (2.93 eV) (Figure 8f). Therefore, the improvement of optical features and energy band structure contributed to highly photocatalytic efficiency. Song, Yang, Chen and Zhang [72] prepared Ag_3_PO_4_/C_60_ photocatalyst via a simple chemical precipitation method. The photodegradation efficiency of MO achieved 93.5% within 8 min of visible light illumination. It is noteworthy that the introduction of C_60_ significantly enhanced the stabilization of Ag_3_PO_4_ which was usually susceptible to photo-corrosion. Additionally, some organic semiconductor nanoparticles composing of fullerene exhibited superior photocatalytic performance. For example, Huo and Zeng [86] successfully fabricated a triphenylamine functionalized bithiazole metal complex hybridized C_60_ photocatalyst. Under visible light irradiation, the photocatalytic H_2_ evolution of the as-prepared photocatalyst showed approximately 4–6-times higher than that of the pristine complex without fullerene. In this photocatalytic system, the organic metal nanocomposite worked as two roles which were both a photosensitizer and a photocatalyst. Additionally, Zhang, et al. [127] prepared an organic photocatalyst of fullerene hydrolyzed aluminum phthalocyanine chloride (AlPc/C_60_) by a reprecipitation method. The photocatalyst showed superior photooxidation degradation of various organic compounds (including N-methyl-2-pyrrolidone (NMP), methanal, and 2-mercaptoethanol). Note that the AlPc/C_60_ exhibited highly efficiency in complete mineralization towards these organic materials, leading to effective CO_2_ generation in reaction solution under visible light irradiation. NMP mineralization experiment was tested in a closed cylindrical reactor containing 10 vol% substrate, wherein the CO_2_ generation amount in AlPc/C_60_ reacting solution reached 3.7 × 10^−7^ mol after 24 h irradiation, which was 2.9-times higher than that of the corresponding C_60_ + AlPc mechanical mixture in solution. It was proposed that this novel photocatalyst based on a biphase structure and features *p*/n junction-like characteristics.

### 4.7. Discussions and Conclusions for Photocatalytic Applications of Fullerene/Semiconductor Photocatalysts

Among fullerene-based photocatalysts, the TiO_2_/fullerene (C_60_ and C_70_) composites have been the most extensively investigated in photocatalytic applications in the past decades. They exhibit efficient performances in wastewater treatment, such as pollutant degradation and disinfection. The synthetic methods are facile and eco-friendly without complicated steps, generally including simple adsorption and hydrothermal synthesis. After fullerene is inserted into TiO_2_, it shows to be fairly helpful in enhancing the photocatalytic efficiency of TiO_2_. However, there is a small deficiency in these materials, namely insufficient utilization of light energy. In other words, they exhibit excellent absorption of UV light but moderate absorption of visible light, while UV irradiation only accounts for 4% in solar irradiation. Besides, metal sulfide/fullerene nanocomposites are an appealing class of photocatalysts for not only decontamination but water splitting for H_2_ generation. They exhibit efficient absorption of visible light together with a moderate stability, such as CdS/C_60_, MoS_2_ and CdS/TiO_2_/C_60_. Compared with the responding pure metal sulfide, they perform significantly boosted efficiency, especially towards H_2_ generation. As to Bi-based semiconductor/fullerene photocatalysts, they appear to only be helpful for pollutant degradation but have not displayed effective capacity for photocatalytic H_2_ generation. There is no doubt that g-C_3_N_4_ has always been one of the hot nanomaterials in photocatalytic area since advent, so g-C_3_N_4_/fullerene photocatalysts are promising options for further photocatalysis. The facile synthetic method makes them attractive materials for photocatalytic applications, such as simple thermal treatment and balling mill. Additionally, it is easy to achieve more intensively oxidation or reduction capacity with tunable bandgap of g-C_3_N_4_.

To sum up, three crucial characteristics of photocatalysts are needed to be considered for wastewater treatment, such as high-efficiency for removing pollutant, stability in duration and nontoxicity to the environment and humans, respectively. In order to convincingly evaluate a photocatalyst, it is required to concern various properties comprehensively, such as optical absorption, energy band, photocatalytic efficiency, stability, cost and so on. As we all know, photocatalytic applications are currently researched in the laboratory, which seldom involves the time consumption of process and economic efficiency. Future work is imperative to focus on these aspects.

## 5. Fullerene/Support (Non-Semiconductor) Photocatalysts for Wastewater Treatment

In addition to fullerene/semiconductor photocatalysts, a series of novel fullerene/solid-support photocatalysts have been developed for wastewater treatment. It is well established that pristine fullerene is extremely insoluble in water (solubility of C_60_ in water < 10^−9^ mg/L), but could dissolve in nonpolar organic solvent, such as toluene and 1,2-dichlorobenzen [128,129]. It is worthwhile mentioning that fullerene solution could induce photochemical reactive oxygen species (ROS) generation via two pathways which were defined as type Ⅰ pathway (Equation (13)) and type Ⅱ pathway (Equation (14)), taking C_60_ as an example as follows [22]. For example, single oxygen ^1^O_2_ can be produced in fullerene-toluene solution (pathway Ⅱ), and O_2_^−•^ and •OH can be generated in solvent in the present of electron donors such as EDTA and NADH (pathway Ⅰ) under UV light illumination [130,131]. While easily aggregation of pristine fullerene in water impedes ROS production owing to self-quenching mechanisms within the aggregates [132,133]. It has been proven that coupling fullerene with hydrophilic functional groups (namely fullerene derivatives) is a helpful strategy to dissolve fullerene in water together with superior ROS generation, such as polyhydroxyl-fullerene, amine-fullerene and other cationic-fullerenes [134,135,136]. At the earlier time, these water-soluble fullerene derivatives were used as photosensitizers for photodynamic therapy, selective antimicrobial and photooxidation organic synthesis [27,137]. More recently, considerable research efforts have been devoted to wastewater treatment for fullerene-support photocatalysts, including photocatalytic pollutant degradation and disinfection in aqueous solution. Without support combination, aqueous fullerene (nC_60_) and fullerene derivatives in aqueous solution are easily decomposed due to photolysis and other external conditions, seriously lowering efficacy of C_60_ as a photocatalyst generating ROS [138,139]. The separation and reutilization are also the barriers of fullerene derivatives used as photocatalysts. Herein, immobilization of fullerene-derivatives on solid support could be a hopeful strategy to fabricate fullerene/solid-support photocatalyst.
(13)C601→hνC60*1→ISCC60*3→O23→O21C601
(14)C601→hνC60*1→ISCC60*3→e−−donorC60−•Ered=+1.14V

Lee, et al. [140] fabricated a series of aminoC_60_/silica photocatalysts by covalent-bond immobilization of aminoC_60_ on 3-(2-succinic anhydride) propyl functionalized silica gel. The synthesis route of the aminoC_60_/silica photocatalysts is presented in Figure 9. In this photocatalytic system, phosphate buffer was required for pollutant degradation and ^1^O_2_ generated by photochemical procedure was the dominating ROS for photocatalytic activity. The photochemical ^1^O_2_ generation ability of the as-prepared photocatalysts were estimated using furfuryl alcohol (FFA) as an indicator and immobilized aminoC_60_ samples exhibited remarkedly higher ^1^O_2_ generation than water-soluble aminoC_60_, among which tetrakis aminoC_60_/silica performed the best (Figure 10a). Accordingly, these aminoC_60_/silica photocatalysts boosted the photocatalytic oxidation degradation towards pharmaceutical pollutants (including ranitidine and cimetidine) as well as photocatalytic disinfection towards MS-2 bacteriophage upon visible light illumination in contrast to corresponding aminoC_60_ alone in aqueous solution. In this case, the immobilization method facilitated well dispersion of C_60_ onto the silica surface, so as to expose more active sites for ROS generation, resulting in enhancement of the photocatalytic efficiency. Additionally, the huge specific surface area of silica significantly enhanced the adsorption of pollutant to the surface of aminoC_60_/silica for promoting closer contact between ^1^O_2_ and the pollutant, because the travel distance of ^1^O_2_ in aqueous is really short within the diffusion length of ∼125 nm over one lifetime [141]. Note that the lifetime of ^1^O_2_ in water was only 3~4 µs which limits the catalytic performance, so immobilization of aminoC_60_ could increase the contact time between ^1^O_2_ and the pollutant from this point of view [142]. In order to further explore the performance of the tetrakis aminoC_60_/silica photocatalyst, a variety of emerging organic contaminants and endocrine disruptors were involved into photocatalytic degradation experiments [143]. The photodegradation rate of ranitidine and propranolol for amino silica/C_60_ were 13.987 ± 0.016 h^−1^ and 10.77 ± 0.019 h^−1^, which were respectively 31-fold and 75-fold faster than that for aimnoC_60_ alone. In particular, the silica/aminoC_60_ was quite effective in trimethoprim degradation while no degradation appeared in aminoC_60_ aqueous solution, which was also observed in a C_60_ aminofullerene-magnetite nanocomposite suspension solution [144]. Moreover, at alkaline conditions (pH 10), acetaminophen, bisphenol A, and 4-chlorophenol could also be effectively degraded over the silica/aminoC_60_ photocatalyst under fluorescent light irradiation. Figure 10b,c compares the photocatalytic efficiency of silica/aminoC_60_ with other semiconductor photocatalysts including TiO_2_, C-TiO_2_ and Pt/WO_3_. It is shown that silica/aminoC_60_ exhibits remarkedly higher ^1^O_2_ generation rate over TiO_2_ and C-TiO_2_ while it performs lower rate than Pt/WO_3_ under fluorescent light or visible light irradiation. Note that the silica/aminoC_60_ exhibits the best efficiency in pharmaceutical (RA and CM) degradation among these materials under visible light irradiation. Although these photoactive catalysts in the comparisons take effect owing to different ROS generation (e.g., primarily ^1^O_2_ upon aminoC_60_, •OH upon TiO_2_ and Pt/WO_3_), it is indicated that silica/aminoC_60_ has a potential for application as an alternative environmental photocatalyst.

There is no denying that aforementioned fullerene/solid photocatalysts are involved into complicated synthesis procedure with fullerene derivatives. Moor and Kim [145] used a simpler method to build solid supported C_60_ photocatalysts via a nucleophilic reaction of a terminal amine onto pristine C_60′_s cage, and through this method SiO_2_/C_60_ and polystyrene resin/C_60_ (PS/C_60_) were developed. The above two photocatalysts both showed higher ^1^O_2_ generation rate than nC_60_ (nanoscale aggregates) in aqueous solution under various illumination conditions, in which the SiO_2_/C_60_ was superior to PS/C_60_. As a photosensitization catalyst, SiO_2_/C_60_ showed effectively photocatalytic MS2 inactivation, which was ascribed to ^1^O_2_-mediated oxidization damage effect. In addition, Moor, Valle, Li and Kim [91] successfully fabricated a MCM-41/C_70_ composite by the same nucleophilic reaction and it was the first time to use C_70_-solid support photocatalyst for wastewater treatment (Figure 11a). Within photoinactivation experiment towards MS2, the MCM-41/C_70_ performed obviously higher efficiency than N-TiO_2_ nanocomposite, which efficiently induced •OH production in aqueous solution for microbial inactivation (Figure 11b,c). It was confirmed that the as-prepared novel MCM-41/C_70_ photocatalyst exhibited efficient photodegradation ability to several pharmaceuticals and personal care products (PPCP), including bisphenol A, 17-α-ethynylestradiol and amoxicillin (Figure 11d,e).

The majority of photocatalysts researched for traditional azo-dye decomposition based upon semiconductor composites. Few studies involved into non-semiconducting materials, especially C_60_/solid support photocatalyst [146]. Whereas, Wakimoto, et al. [147] prepared a SiO_2_/C_60_ powder via a simple adsorption using pristine C_60_ and silica gel in toluene and it was important to highlight that a novel route for dye photodegradation was proposed for the C_60_/solid support photocatalyst. Unlike typically semiconductor photocatalysts, the SiO_2_/C_60_ composite exhibited effectively visible-light-driven photocatalytic degradation towards methyl orange in the presence ascorbic acid, while SiO_2_/C_60_ alone without ascorbic acid did not show degradation performance. In this case, ascorbic acid could protonate MO and transform it into quinoid form with strong electron-acceptability. It was proved that both ^1^O_2_ and O_2_^•−^ species took part in the dye degradation. On one hand, C_60_ dispersed onto silica surface was excited to undergo the intersystem crossing from the single to triplet state for ^1^O_2_ generating, then the generated ^1^O_2_ could attack rich-electron quinoid structure to decompose MO. On the other hand, the electron transferred from ascorbic acid to the excited C_60_ to forming the C_60_ radical anion, followed by generation of O_2_^•−^ via O_2_ receiving the electron from C_60_^•−^, wherein O_2_^•−^ was an effective specie for dye degradation. It was also confirmed that methyl red could be decomposed by SiO_2_/C_60_ at the same conditions. Likewise, Kyriakopoulos, et al. [148] successfully fabricated a MCM-41/C_60_ photocatalyst via a dry impregnation method. Coupling C_60_ with MCM-41 significantly increased the BET of the as-prepared photocatalyst, thus effectively dispersing C_60_ clusters as well as strengthening its adsorption ability to pollutant. The optimum 3MCM-41/C_60_ (3 wt % C_60_) sample showed 74.9% decolorization efficiency in Orange G, which was markedly higher than that of C_60_ alone. This photocatalyst proved to be remarkably stable, wherein less than 5% photodegradation efficiency was lost after five cycles.

Fullerene-based solid photocatalysts could effectively prevent fullerene aggregations and enhance the photo-stabilization of fullerene alone as well as enhance the pollutant adsorption due to the introduced support, thus leading to increase the photocatalytic efficiency. In the meanwhile, this fullerene-based photocatalyst could perform selected oxidization of pollutant with ^1^O_2_ that can prevent natural organic matter (NOM) interference, which underline the potential of these materials for wastewater treatment in natural water. Therefore, fullerene-based support photocatalysts are promising materials for environmental applications and further effort will be required to fabricate novel photocatalysts of this type.

## 6. Conclusions and Perspectives

In summary, the essence of photocatalysis bases on the ROS generation in the presence of light resource irradiation. Fullerenes, including C_60_ and C_70_, have been extensively investigated in the photocatalytic application due to their unique optical and photochemical characteristics. Fullerene could be anchored on semiconductors to enhance their photocatalytic activity, and also supported on non-semiconductor solids to fabricate novel fullerene-based photocatalysts due to its self-photocatalytic features. In the present review, fullerene/semiconductor photocatalysts and fullerene-solid support photocatalysts are summarized for wastewater treatment (pollutant degradation, Cr(VI) reduction, disinfection etc.) and water splitting for H_2_ generation. A number of synthesis methods have been used to fabricate semiconductor/fullerene photocatalysts, including simple adsorption, hydrothermal synthesis, ball milling, sol-gel and so on. The semiconductors alone usually display limited photocatalytic performance, wherein the fast recombination of photoinduced hole-electron pairs and inefficient light energy utilization are the two main obstacles. Whereas, fullerene could availably enhance the photocatalytic efficiency of semiconductors by retarding the recombination of hole-electron pairs and increasing the light absorption (UV and visible light). In some cases, semiconductor/fullerene photocatalysts display better stabilization than semiconductors alone, and the introduced fullerene increases the BET of the semiconductor for enhancing the pollutant adsorption. The studies manifest that excess fullerene inhibits the photocatalytic ability due to the coverage effect towards excited sites. So, a suitable amount of fullerene is imperative for superior photocatalytic activity of semiconductor/fullerene photocatalysts. Plentiful semiconductors have been coupled with fullerene for wastewater treatment and water splitting, such as TiO_2_, ZnO, CdS and C_3_N_4_. The photocatalytic effect of these photocatalysts are presented and the involved mechanisms are discussed in detail in this review, including the reaction of ROS generation and the transfer route of electron. On the other hand, fullerene-solid support photocatalysts are also discussed in application for wastewater treatment, such as silica/C_60_, MCM-41/C_70_ and polysiloxane-supported fullerene photocatalyst. They display excellent photoinduced ROS (mainly ^1^O_2_) generation in aqueous solution after fullerene was dispersed onto the solid support, which is the direct factor contributing to the photocatalytic reaction. Meanwhile, they effectively enhance the photo-stabilization of fullerene alone as well as enhance the pollutant adsorption. It is noteworthy that these fullerene-based photocatalysts perform selected oxidization of the pollutant, wherein the photoinduced ^1^O_2_ could prevent natural organic matter (NOM) interference. So, it underlines the potential of these materials for wastewater treatment in natural water.

Although some encouraging properties have been achieved for fullerene-based photocatalysts, the development of fullerene-based photocatalysts still has remaining challenges. (1) The interface contact between the fullerene and semiconductor is not so intimate, leading to limiting the electron transport ability and photostability of semiconductor/fullerene photocatalysts. The majority of semiconductor/fullerene photocatalysts formed by simple adsorption and hydrothermal synthesis, while few studies confirmed the existence of covalent bond or other tight bond in them. (2) The photocatalytic mechanisms of fullerene/semiconductor photocatalysts are partly not clear. Some studies indicated that the electrons transfer from the semiconductor to the fullerene due to the strong electron-accepting ability of fullerene, while others deemed that the electrons transfer in the opposite direction. (3) Not enough research has been done on the novel fullerene/solid (non-semiconductor) photocatalysts, in which the selection of solid support is restricted to silica, MCM-41 and polysiloxane. (4) The majority of fullerene-based photocatalysts are investigated in the treatment of simulated wastewater with the artificial addition of a single pollutant, and actual industrial wastewater is rarely involved.

In our opinion, several directions are worthy of attention for fullerene-based photocatalysts in the future: (1) Innovative strategy should be developed to construct semiconductor/fullerene photocatalysts with efficient performance and high stability. (2) Further works should focus on mechanism studies of the semiconductor/fullerene composite, especially the electron-transfer path which is still in dispute. (3) More attention should be paid to fullerene derivatives, which are promising materials for developing novel fullerene-based photocatalysts. (4) The studying of fullerene-based photocatalysts on their actual performance in natural water, industrial wastewater and multi-polluted wastewater.

## Figures and Tables

**Figure 1 materials-13-02924-f001:**
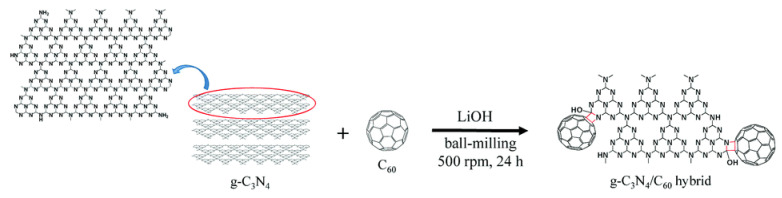
Schematic illustration of the mechanochemical reaction between g-C_3_N_4_ and C_60_ in the presence of the LiOH catalyst in a sealed ball-mill crusher. Reproduced with permission from Reference [55]. Copyright 2017, RSC.

**Figure 2 materials-13-02924-f002:**
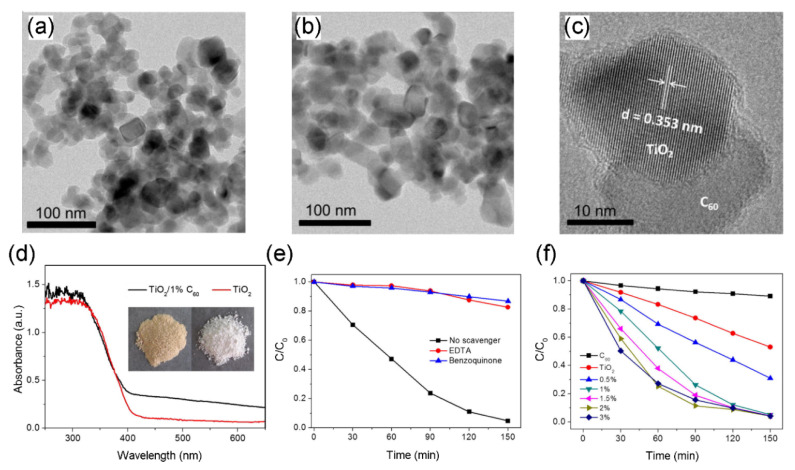
TEM images of TiO_2_ (**a**) and TiO_2_/C_60_ (**b**). (**c**) HR-TEM image of TiO_2_/C_60_. (**d**) Diffuse reflectance spectroscopy (DRS) of TiO_2_/C_60_ and pure TiO_2_. (**e**) Free radical capture experiment within photocatalytic degradation of RhB. (**f**) Photocatalytic degradation towards RhB over the TiO_2_/fullerene nanocomposite under the visible light irradiation. Reproduced with permission from Reference [68]. Copyright 2016, Elsevier.

**Figure 3 materials-13-02924-f003:**
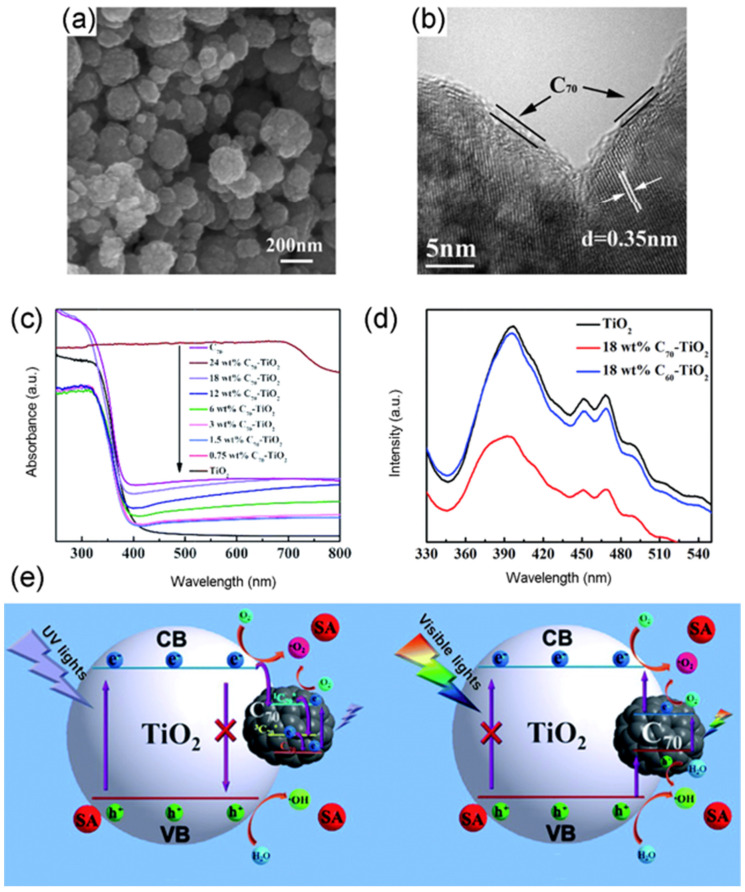
SEM (**a**) and TEM (**b**) images of 18 wt % C_70_-TiO_2_. (**c**) UV-Vis DRS of the C_70_-TiO_2_ and pure TiO_2_. (**d**) Comparison of PL spectra over C_70_-TiO_2_, C_60_-TiO_2_ and TiO_2_. (**e**) Photocatalytic mechanisms of C_70_-TiO_2_ under UV and visible light illumination. Reproduced with permission from Reference [69]. Copyright 2015, RCS.

**Figure 4 materials-13-02924-f004:**
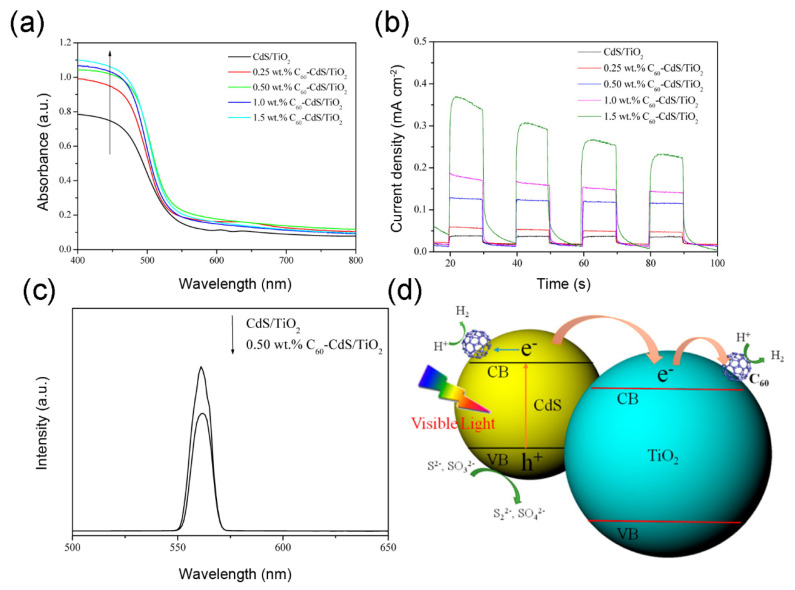
UV-vis spectra (**a**), photocurrent density measured at 0.5 V in a 0.5 M aqueous Na_2_SO_4_ electrolyte (**b**) and PL spectra (**c**) excited by 280 nm of CdS/TiO_2_ and C_60_-CdS/TiO_2_ nanocomposites. (**d**) The mechanism of photocatalytic H_2_ generation over C_60_-CdS/TiO_2_. Reproduced with permission from Reference [84]. Copyright 2015, ACS.

**Figure 5 materials-13-02924-f005:**
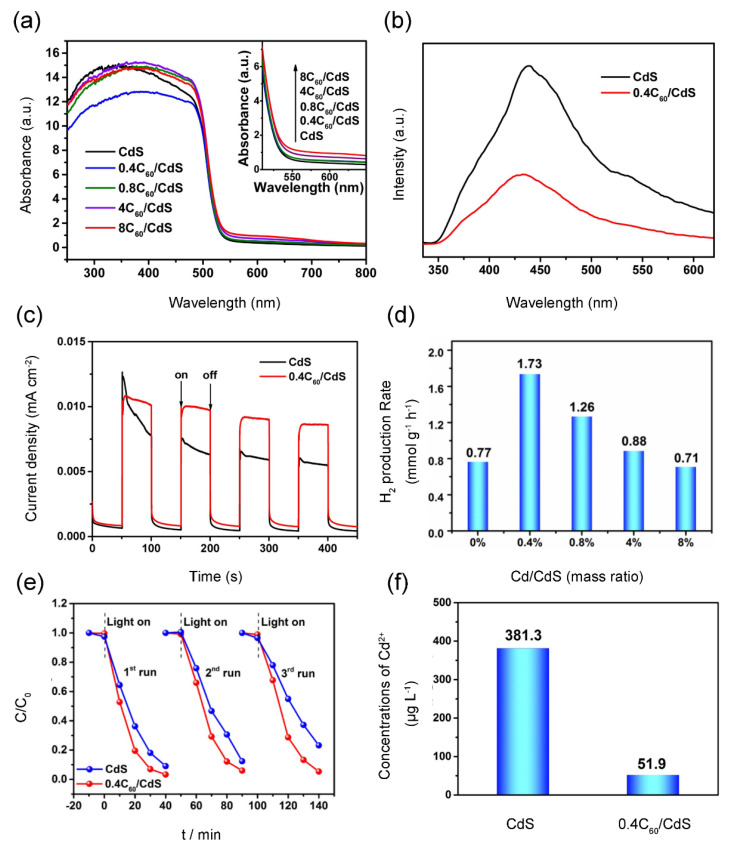
UV-vis spectra (**a**), PL emission spectra (**b**) and transient photocurrent responses (**c**) in 0.5 M Na_2_SO_4_ solution of CdS and C_60_/CdS nanocomposites. (**d**) The photocatalytic rate of H_2_ generation over C_60_/CdS samples under visible light illumination. (**e**) Recyclability test of photodegradation towards RhB under visible light illumination over CdS and 0.4C_60_/CdS composite. (**f**) Comparison of Cd^2+^ concentrations in the solutions of CdS and 0.4C_60_/CdS photocatalysts after three cycles for RhB degradation. Reproduced with permission from Reference [43]. Copyright 2017, Elsevier.

**Figure 6 materials-13-02924-f006:**
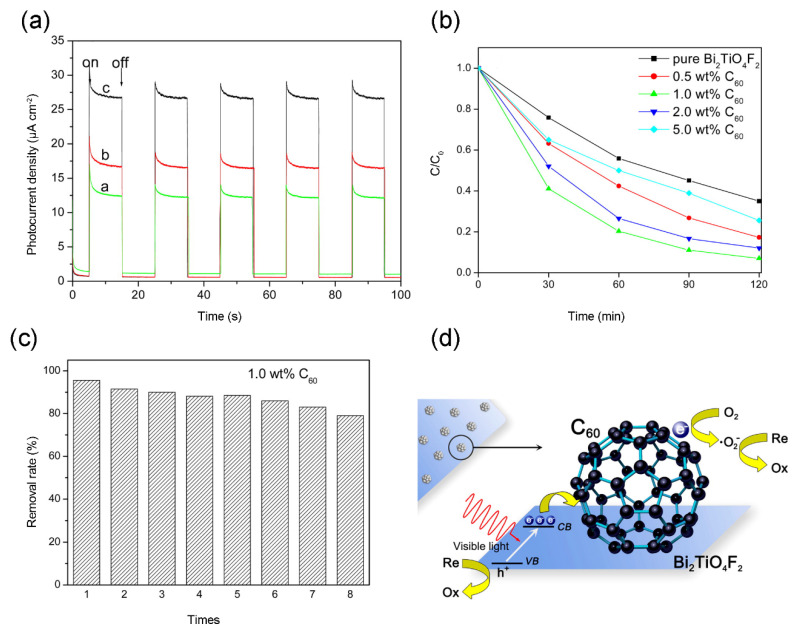
(**a**) Photocurrent responses of Bi_2_TiO_4_F_2_, C_60_ (1 wt %) + Bi_2_TiO_4_F_2_ mixture and 1 wt % C_60_/Bi_2_TiO_4_F_2_. (**b**) Photocatalytic performance towards RhB degradation. (**c**) Recyclability test of the as-prepared composites 1 wt % C_60_/Bi_2_TiO_4_F_2_. (**d**) The mechanism of C_60_/Bi_2_TiO_4_F_2_ photocatalyst is presented under visible light irradiation. Reproduced with permission from Reference [73]. Copyright 2013, ACS.

**Figure 7 materials-13-02924-f007:**
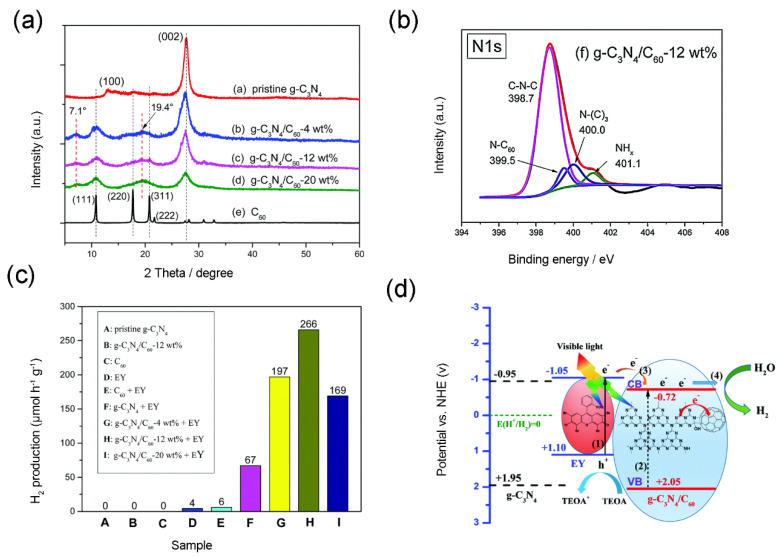
(**a**) XRD patterns of g-C_3_N_4_/C_60_ samples and pristine g-C_3_N_4_. (**b**) High-resolution N 1s XPS spectra of g-C_3_N_4_/C_60_−12 wt % nanocomposite. (**c**) Photocatalytic H_2_ generation rates of the as-prepared samples. (**d**) A schematic of the photocatalytic H_2_ generation mechanism for the g-C_3_N_4_/C_60_ nanocomposite. Reproduced with permission from Reference [55]. Copyright 2017, RCS.

**Figure 8 materials-13-02924-f008:**
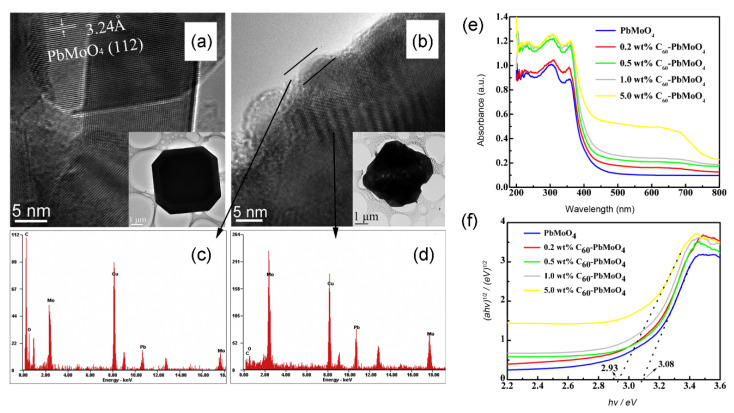
HRTEM images of PbMoO_4_ (**a**) and C_60_-PbMoO_4_ (**b**). EDS spectrum measured from the edge (**c**) and the center of C_60_–PbMoO_4_ composite (**d**). (**e**) DRS of the C_60_–PbMoO_4_ samples and pure PbMoO_4_ composite. (**f**) Plot of (αhν)^1/2^ versus photon energy (hν) based on the DRS. Reproduced with permission from Reference [50]. Copyright 2013, Elsevier.

**Figure 9 materials-13-02924-f009:**
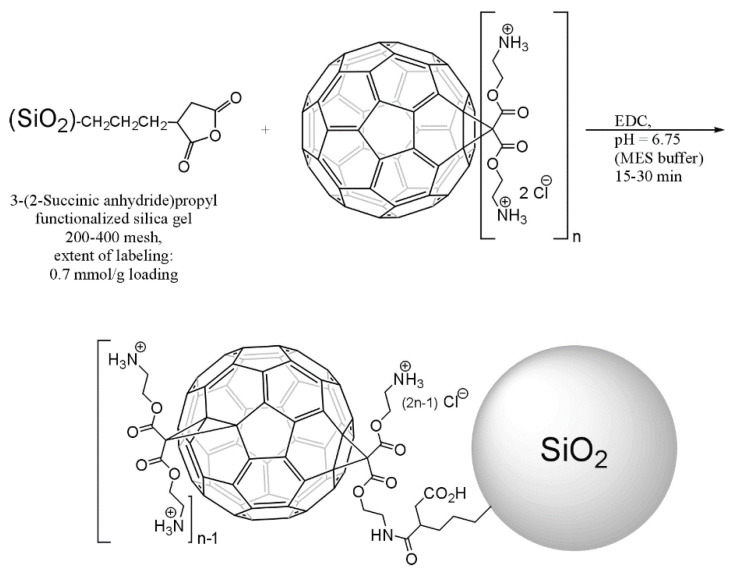
Route for immobilization of aminofullerenes on functionalized silica gel. Reproduced with permission from Reference [140]. Copyright 2010, ACS.

**Figure 10 materials-13-02924-f010:**
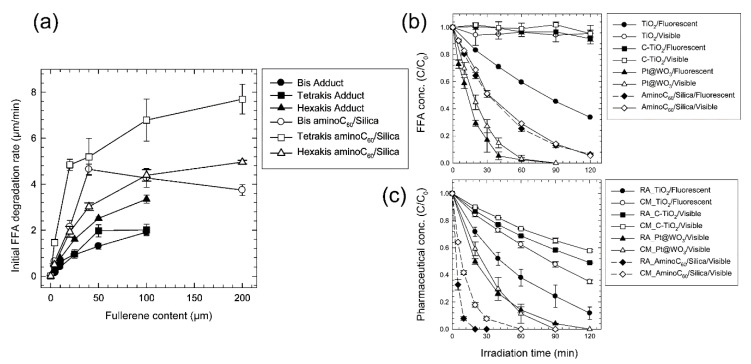
(**a**) Photochemical furfuryl alcohol (FFA) degradation (measuring photosensitized ^1^O_2_ production) by aminofullerenes and aminofullerene/silica composites. Comparisons of degradation efficiency towards furfuryl alcohol (**b**) and ranitidine (RA) and cimetidine (CM) (**c**) by TiO_2_, carbon-doped TiO_2_ (C-TiO_2_), Pt@WO_3_ and tetrakis aminoC_60_/silica. Reproduced with permission from Reference [140,143]. Copyright 2010, 2011, ACS.

**Figure 11 materials-13-02924-f011:**
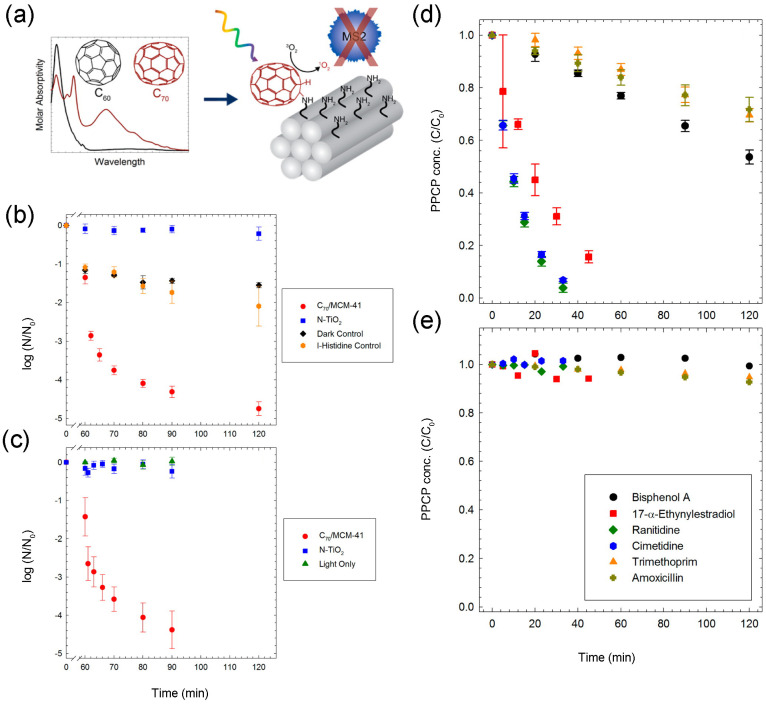
(**a**) A schematic of photocatalytic mechanism of novel C_70_/MCM-41 photocatalyst. Photoinduced MS2 inactivation kinetics of C_70_/MCM-41 and porous N-TiO_2_ under visible light (**b**) and sunlight irradiation (**c**). Photodegradation kinetics of various PPCPs by C_70_/MCM-41 under visible irradiation (**d**) and corresponding dark controls (**e**). Reproduced with permission from Reference [91]. Copyright 2015, ACS.

**Table 1 materials-13-02924-t001:** Summary of fullerene based photocatalysts for pollutant degradation.

Photocatalyst (Additive Amount)	Synthesis Method (Fullerene Content)	Pollutants	Experimental Conditions (Light Source, Pollutant Concentration and React Time)	Photocatalytic Activity	Enhancement Factor	Reference
TiO_2_/C_60_ (1 g/L)	In-situ growth (2.0 wt %)	Methylene blue (MB)	UV irradiation, 1.0 × 10^−4^ mol/L, 60 min	99%	around 75% for TiO_2_	[67]
TiO_2_/C_60_ (1 g/L)	Ultrasonication–evaporation (1.0 wt %)	RhB	500 W Xe-lamp (>400 nm), 10 mg/L, 150 min	95%	below 5% for TiO_2_	[68]
TiO_2_/C_70_ (1 g/L)	Hydrothermal synthesis (8.5 wt %)	Sulfathiazole	300 W Xenon lamp (>420 nm), 10 mg/mL, 180 min	80%	10% for TiO_2_	[69]
ZnO/C_60_ (0.5 g/L)	Simple adsorption (1.5 wt %)	MB	8 W UV lamp (λ = 254 nm), 8 mg/L	k = 0.0569 min^−1^	3-times than ZnO	[70]
ZnO/C_60_ (0.83 g/L)	Chemical vapor (16.7 wt %)	Phenol	1500 W xenon lamp simulating solar light, 20 mg/L	k = 0.160 min^−1^	1.22-times than ZnO	[42]
ZnFe_2_O_4_@C_60_ (1 g/L)	Hydrothermal synthesis	Norfloxacin	Solar irradiation, 20 mL of 50 ppm norfloxacin, 90 min	85%	60% for ZnFe_2_O_4_	[71]
WO_3_@C_60_	Hydrothermal synthesis (4.0 wt %)	MB	Visible light, 90 min	94%	Inferior degradation efficiency for pure WO_3_	[53]
ZnAlTi-LDH@C_60_ (ZnAlTi-LDO) 0.5 g/L	Precipitation (5%)	Bisphenol A (BPA)	300 W xenon lamp simulating visible light, 10 mg/L, 60 min	80%	below 10% for ZnAlTi-LDH	[38]
CdS/C_60_ (1 g/L)	One-pot hydrothermal method (0.4 wt %)	RhB	300 W xenon lamp (>420 nm), 20 mL, 10 ppm of RhB	k = 0.089 min^−1^	1.5-times than CdS	[43]
C_3_N_4_/C_60_ (0.6 g/L)	Simple adsorption (1.0 wt %)	RhB	500 W xenon lamp (>420 nm), 50 mL, 1.0 × 10^−5^ mol l^−1^ RhB, 60 min	97%	54% for C_3_N_4_	[45]
g-C_3_N_4_/C_60_ (0.5 g/L)	Calcination (0.03 wt %)	MB, phenol	500 W xenon lamp (>420 nm), MB (50 mL, 0.01 mM), phenol (50 mL, 5 ppm).	k_1_ = 1.036 h^−1^, k_2_ = 0.093 h^−1^	3.2- and 2.9-times than C_3_N_4_	[35]
Ag_3_PO_4_/C_60_ (0.5 g/L)	Precipitation (2.0 wt %)	Acid red 18 (AR18)	400 W halogen lamp (420–780 nm, 21.5–23.0 mW cm^−2^), 50 mL, 6.5 × 10^−5^ mol/L of AR18, 60 min	90%	53% for Ag_3_PO_4_	[31]
Ag_3_PO_4_/C_60_ (1 g/L)	Precipitation (5.0 mg/L)	Methyl Orange (MO)	300 W xenon lamp (>420 nm), 10 mg/L	k = 0.453 min^−1^	k = 0.028 min^−1^ for Ag_3_PO_4_	[72]
PbMoO_4_/C_60_ (0.4 g/L)	Hydrothermal synthesis (0.5 wt %)	RhB	18 W low-pressure mercury lamp as the UV light source, 50 mL of RhB (1 × 10^−5^ M), 2 h	99%	37% for PbMoO_4_	[50]
Bi_2_WO_6_/C_60_ (1 g/L)	Simple adsorption (1.25 wt %)	MB, RhB	500 W xenon lamp (>420 nm), 1 × 10^−5^ mol/L RhB or MB (100 mL)	k_1_ = 0.0099 min^−1^, k_2_ = 0.0454 min^−1^	5.0- and 1.5-times than Bi_2_WO_6_	[40]
BiOCl/C_70_ (1 g/L)	In-situ growth (1.0 wt %)	RhB	500 W xenon lamp (>420 nm), 10 mg/L, 30 min	99.8%	49.7% and 66.4% for BiOCl and P25 (TiO_2_)	[39]
Bi_2_TiO_4_F_2_/C_60_	Solvothermal method (1.0 wt %)	RhB	Visible light, 20 ppm RhB, 120 min	93%	65% for Bi_2_TiO_4_F_2_	[73]
CNTs/BiVO_4_-C_60_ (2 g/L)	Hydrothermal synthesis (2.5 wt %)	RhB	300 W xenon lamp (>420nm), 100 mL, 0.01 mmol/L RhB, 30 min	96.1%	74.0% for BiVO_4_	[51]
CNTs/Bi_2_MoO_6_-C_60_ (2 g/L)	Hydrothermal synthesis (2.5 wt %)	RhB	300 W xenon lamp (>420 nm), 100 mL, 0.01 mmol/L RhB, 30 min	88.4%	43.7% for Bi_2_MoO_6_	[51]
Ag_3_PO_4_/Fe_3_O_4_/C_60_ (1 g/L)	Hydrothermal synthesis (5.0 wt %)	MB	400 W mercury lamp (>420 nm), 50 mL of MB (25 mg/L), 300 min	95%	33% for Ag_3_PO_4_	[74]
TiO_2_/Pt-C_60_ (1 g/L)	Sol-gel method (7.5 wt %)	MO	8 W halogen lamp (400–790 nm), 50 mL, 1 × 10^−5^ mol/L of MO	k = 3.67×10^−3^ min^−1^	1.58- and 16.4-times than Pt/TiO_2_ and TiO_2_	[75]
TiO_2_/Pd-C_60_ (1 g/L)	Sol-gel method (21 wt %)	MB	UV lamp box (8 W, 365 nm), 50 mL, 1 × 10^−4^ mol/L of MB	k = 0.0337 min^−1^	14-times than TiO_2_	[76]
Au/TiO_2_-C_60_ (1 g/L)	Hydrothermal synthesis (3.25 wt %)	MO	500W tungsten halogen lamp, 20 mL, 10 mg/L of MO, 160 min	95%	47% for TiO_2_	[77]
TiO_2_/CdS-C_60_ (1 g/L)	Sol-gel method (5.0 wt %)	MB	8 W halogen lamp (400–790 nm), 50 mL, 1 × 10^−5^ mol/L of MB	k = 7.9×10^−3^ min^−1^	4.9- and 3.5-times than CdS/TiO_2_ and TiO_2_	[56]
TiO_2_/WO_3_-C_60_ (1 g/L)	Sol-gel method (3.0 wt %)	MO	8 W halogen lamp (400–790 nm), 50 mL, 1 × 10^−5^ mol/L of MO	k = 4.75×10^−3^ min^−1^	1.66- and 21.2-times than WO_3_/TiO_2_ and TiO_2_	[78]
TiO_2_/CD/C_60_ (1 g/L)	Simple adsorption (1.5%)	MB, 4-chlorophenol (4-CP)	84 W light sources (>420 nm), MB (10 mL, 144 μM), 10 mg/L 4-CP	k_1_ = 0.014 min^−1^, k_2_ = 0.036 min^−1^	2- and 4.9-times than TiO_2_	[79]
TiO_2_/Fullerol (1 g/L)	Wet impregnation	Procion red MX-5B	16 solar UVA lamps (350 nm)	k = 0.0128 min^−1^	2.6-times than TiO_2_	[80]
TiO_2_/Fullerol (1 g/L)	Wet impregnation (1.0 wt %)	Formic acid	Hg lamp (365 nm)	k = 91.0 µmol L^−1^ min^−1^	1.3-times than TiO_2_	[59]
Nb-TiO_2_/Fullerol (0.5 g/L)	Simple adsorption	4-chlorophenol	300-W Xe arc lamp (>420 nm)	k = 13.9×10^−3^ min^−1^	3.3-times than P25	[81]

k means rate constant of photocatalytic degradation, which is calculated by the relationship between−ln(C/C_0_) and t (C_0_ and C are the concentrations of pollutant in solution at times 0 and t, respectively).

**Table 2 materials-13-02924-t002:** Summary of fullerene based photocatalysts for photocatalytic H_2_ generation.

Photocatalyst (Additive Amount)	Synthesis Method (Fullerene Content)	Experimental Conditions	Photocatalytic Rate of H_2_ Generation	Enhancement Factor	Reference
CdS/C_60_ (0.5 g/L)	Hydrothermal synthesis (0.4 wt %)	300 W xenon lamp (>420 nm), 50 mL aqueous solution containing 10 vol% lactic acid and 1 wt % Pt	1.73 mmol h^−1^ g^−1^	2.3 Times of pure CdS	[43]
WO_3_@C_60_ (0.5 g/L)	Hydrothermal synthesis (4 wt %)	300 W xenon lamp (>420 nm), Triethanolamine (TEA)	154 µmol h^−1^ g^−1^	2 times of pure WO_3_	[53]
MoS_2_/C_60_ (0.5 g/L)	Ball milling method (2.8 wt %)	300 W xenon lamp (>420 nm), 20 mL aqueous solution containing 3.5 mg Eosin Y (EY) and 1 mL TEA	6.89 mmol h^−1^ g^−1^	9.3 times of ball-milled MoS_2_	[54]
g-C_3_N_4_/C_60_ (1 g/L)	Ball milling method (12 wt %)	300 W xenon lamp (>420 nm), 100 mL aqueous solution containing 17.5 mg EY and 5 mL TEA	266 µmol h^−1^ g^−1^	4.0 times higher than pristine C_3_N_4_	[55]
Cr_1.3_Fe_0.7_O_3_-C_60_ (5 mg/78 mL)	Simple adsorption (3%)	300 W xenon lamp (>420 nm), 78 mL 10 vol% TEA aqueous solution	220.5 µmol h^−1^ g^−1^	2 times of the Cr_1.3_Fe_0.7_O_3_	[82]
Fe_2_O_3_/C_60_ (5 mg/78 mL)	Simple adsorption (0.5~1 wt %)	300 W xenon lamp (>420 nm), 78 mL 10 vol% TEA aqueous solution	β-Fe_2_O_3_/C_60_: 1665 µmol h^−1^ g^−1^; α-Fe_2_O_3_/C_60_: 202.9 µmol h^−1^ g^−1^; γ-Fe_2_O_3_/C_60_: 169.4 µmol h^−1^ g^−1^	β-Fe_2_O_3_: 169.4 µmol h^−1^ g^−1^; α-Fe_2_O_3_: 80.6 µmol h^−1^ g^−1^; γ-Fe_2_O_3_: 252 µmol h^−1^ g^−1^; C_3_N_4_: 82.7 µmol h^−1^ g^−1^	[83]
CdS/TiO_2_-C_60_ (50 mg/80 mL)	An ion-exchanged method (0.5 wt %)	Low power UV-LEDs (420 nm), 80 mL solution (0.25 M Na_2_S, 0.25 M Na_2_SO_3_)	120.6 µmol h^−1^ g^−1^	8.5 times of CdS/TiO_2_	[84]
TiO_2_/C_60_-d-CNTs (1 g/L)	Hydrothermal synthesis (5 wt %)	300 W xenon lamp (>420 nm), 100 mL 10 vol% TEA aqueous solution	651 µmol h^−1^ g^−1^	208 µmol h^−1^ g^−1^ for pure TiO_2_	[85]
g-C_3_N_4_/graphene/ C_60_ (2 g/L)	Wet impregnation	Light-emitting diode (>420 nm), 50 mL solution containing 1 wt‰ Pt and 10 vol% TEA	545 µmol h^−1^ g^−1^	50.8 and 4.24 times of graphene/g-C_3_N_4_ and C_60_/g-C_3_N_4_	[19]
(2TPABTz)–metal complex/C_60_	Simple adsorption (2 wt %)	300 W xenon lamp (>420 nm), an aqueous lactic acid (LA)	2TPABTz-Cu/C_60_: 4.05 mmol h^−1^ g^−1^; 2TPABTz-Co/C_60_: 3.77 mmol h^−1^ g^−1^; 2TPABTz-Ru/C_60_: 6.12 mmol h^−1^ g^−1^	2TPABTz-Cu: 4.05 mmol h^−1^ g^−1^; 2TPABTz-Co: 3.77 mmol h^−1^ g^−1^; 2TPABTz-Ru: 6.12 mmol h^−1^ g^−1^; TiO_2_ (P25): 0.072 mmol h^−1^ g^−1^	[86]
WO_3_/C_60_@Ni_3_B/Ni(OH)_2_ 2 g/L	Photo-deposition technique	500 W xenon lamp (>420nm), 100 mL 10 vol% TEA aqueous solution	1.578 mmol h^−1^ g^−1^	9.6 times of pure photocatalyst	[87]

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
