# Peer review of "Recent Progress on Fullerene-Based Materials: Synthesis, Properties, Modifications, and Photocatalytic Applications"

_materials, 2020, doi:10.3390/ma13132924_

Round 1

Reviewer 1 Report

Referee report on Manuscript ID: materials-816619
Title: Recent Progress on Fullerene-Based Materials: Synthesis, Properties, Modifications, and Photocatalytic Applications.
Authors: Sai Yao, Xingzhong Yuan *, Longbo Jiang, Ting Xiong, Jin Zhang.

Submitted to section: Carbon Materials

Subsection: Advanced Carbon Materials For Catalytical Applications

 The theme of this review (photocatalytic applications, such as water splitting, Cr (Ⅵ) reduction, pollutant degradation and bacterial disinfection) is important for the global goal of human beeing regarding to enviromental safety and improvement of ecological situation on the Earth.  Authors discussed a large amount of researches dedicated to photocatalic process with the focus on perspective application of composite materials containing fullerenes and fullerene derivatives.

For the review article I would recommend to insert some more clear and simple explanation about the sence of photocatalisys. Then describe the similarities and differences in the photocatalysis of discussed processes in the article.  For example: wastewater treatment processes and the photocatalysis in the process of hydrogen production by splitting the water.

Explain the reason and importance of Cr (VI) reduction process. Is this prescribed by some European derictives? Why this kind of researchs are important?

I would recommend to insert the meanings of some abbreviations to be clear for readers that are not  familar well with topic (See comments for improvement of the article below).

For the Table 1 and Table 2 I would recommend to give more decription about considered content of Tables. For example, give brief explanation about photocatalytic activity. How it was measured and some comparison of such parameters like 60 min in 99% for TiO2/C60 and/or 0.0569 min-1 in the case of ZnO /C60. As I understood the min and min-1are probably different mearsures. If you have a review it would be nice to be clear for readers not familar with topic in detailes. This definitions  (information) can be given in broad sence. The same is related to explanation of meaning »Enhancement factor« in the tables in broad sence.

Then it would ne nice to make conclusions to the information given in the Table 1 and 2.

Which of composites are the most perspective? Which factors are the most important in the sence of time consumption of process and from econimical point of view as well as their effectiveness.

Which of considerd composites represented the best  results and can be recommended for treatment of water polution? What characteristics are the most crucial  (important)?

Comments for improvement the article.

Page 2, line 80. What does the abbreviations »EDTA and NADH« means? Please insert in the text.

nicotinamide adenine dinucleotide (NADH)?

ethylenediamine tetraacetic acid (EDTA)?

Page 3, line 142. What does it mean BET?

Is it Brunauer, Emmett and Teller (BET)?

Page 5, line 212. Give full name for abbreviation vdW. Is it Van Der Waals?

Page 6, line 251. Insert the point at the end.

Page 7. Table 1. Please indicate somethere the meaning of abbreviation MB. Is it methylene blue (MB)? MO-?

Please indicate the meaning of abbreviation BPA. Is it Bisphenol A (BPA)

Table 2. The abbreviations such as »EY, TEOA...« should be explaned.

Page 12, linnes 293-295. It is not clear explanation concerned the Figure 2e

»As depicted in Fig. 2e, the photoinduced hole (h+) and superoxide radical (•O2-) were involved in the photocatalytic reaction«.  Please, clarify. It is not shown in the Fig 2e but it probably can be concluded.

Page 13, line 312. Please, clarify the abbreviation »CTAB«.

Page 13, line 316. »iO2-C70nanocomposites«- insert space between iO2-C70 and nanocomposites.

Page 13, line 343. What is MCPBA? Is it meta-Chloroperoxybenzoic acid (mCPBA)?

  I would recommend the manuscript for publishing after corrections.

Reviewer 2 Report

This review summarizes fullerene-based materials (including fullerene/semiconductor and fullerene/non-semiconductor photocatalysts) for photocatalytic applications, such as water splitting, Cr (Ⅵ) reduction, pollutant degradation and bacterial disinfection. The manuscript is well organized, so I would like to ask the authors to revise the following minor points. 

How about the materials' stability? This point is very critical if we consider the practical applications.   The presented data are reasonable/standard data in similar research. In addition, it is not very general and deep comparisons. More comments on this point are helpful, perhaps previously reported papers should be cited as the refs.   Related papers have been reported by different research groups. It is better to cite the following refs to support some related paragraphs in the introduction part. 

Langmuir 30 (2), 651-659, 2014 -- TiO2  

Overall the manuscript is well written, but I want to see the authors' perspective (future vision) on this research in the conclusion part.   Some little typo errors are found. Please carefully check the sentences again.   The presented figures should be re-arranged. The font sizes, and font types should consistent through the manuscript.   

Reviewer 3 Report

This review describes photocatalytic applications of C60 and related compounds.

No problems could be found, so it should be accepted as it is.

Please confirm this point at the proof stage.

L303 p-electron -> pi-electron (?)

That's all.

Reviewer 4 Report

Dear authors,
This paper concerns a review of what has been doing in the area of fullerene. However, I do not see the major contributions of it in the field of materials.
My general comments are:
It would be helpful if the authors do some rearrangements in the text. There are a lot of mistakes in the text, namely grammar and typos errors, affirmations without support to references, indifference to the terms "adsorption" and "absorption", and some figure need to be changed.
My specific comments are:
Please, find below my comments, which should be taken in consideration.
In introduction section, the authors did not mention the main obstacles in using fullerene in water treatment applications since they can suffer from aggregation behavior when dispersed in water, forming nanoscale aggregates (termed nC60, nC70, etc.). Thus, the lack of significant photoactivity mentioned by the authors will be from the quenching of excited states of neighboring fullerene molecules which are brought into close contact via aggregation. For retaining fullerene’s photoactivity in aqueous systems it is necessary the immobilization of fullerene onto solid supports.
Although, in lines 295-296 the authors said and I cite: "Under visible light illumination for 150 min, 1wt% TiO2/C60 nanocomposite showed 95% 295 degradation efficiency of RhB, which was significantly higher than pristine TiO2." For me, it is necessary to show the graph corresponding to the photoactivty of pristine TiO2.
In lines 304-305 the authors said and I cite: " Similar to C60, C70 has higher electron acceptability and higher efficiency of light harvesting over C60 [80,81]." This is a mistake! The authors should compare fullerene with TiO2. Thus, the correct sentence is: Similar to C60, C70 has higher electron acceptability and higher efficiency of light harvesting over TiO2 [80,81].
Line 719 is missing.
The conclusion is well organized.
Meanwhile, the following figures must be corrected:
Fig. 2(c) should be corrected since the vertical axis is not coherent with other graphs (Absorbance must include a.u. term).
Fig. 3(c) should be changed since the colors used for each sample are too similar and this makes the figure unclear.
Fig. 3(d) has designed Intensity vs wavelength. I do not understand why the authors have changed from Absorbance to Intensity. They should use the same quantity.
Fig. 6(a) has to be emended since the vertical axis is not written properly (Photocurrent density (μA cm-2).
Fig. 7(a-c) units of the variables must be in parenthesis as an authors' option.
Fig. 8(a-f) should be changed. The letters are too big and the graphs (e) and (f) have the vertical axis without units or units written in the wrong format (the authors must be homogeneous in between the figures)
Fig. 9(b-d) should be changed and resolution increased. The vertical axis has quantities written in the wrong format. For example: Fig. 9(b) says FFA Conc. and it should be FFA conc. the same is seen for the other graphs.
Fig. 10 has the same problems as the Fig. 9.
The graphical abstract does not correspond to the work done. It should be improved.
The reviewer,

Round 2

Reviewer 4 Report

The authors made a great effort to satisfy the reviewer's suggestions, and, therefore, the manuscript was improved. However, no significant contribution in the field is being made with this publication, in my humble opinion. I leave the decision of what is good for the magazine's readers.